# THE LLM BOTTLENECK: WHY OPEN-SOURCE VISION LLMS STRUGGLE WITH HIERARCHICAL VISUAL UNDERSTANDING

## ABSTRACT

This paper reveals that many open-source language models (LLMs) lack hierarchical knowledge about our visual world, unaware of even well-established biology taxonomies. This shortcoming makes LLMs a bottleneck for vision LLMs' hierarchical visual understanding (e.g., recognizing `Anemone Fish` but not `Vertebrate`). We arrive at these findings using about one million four-choice visual question answering (VQA) tasks constructed from six taxonomies and four image datasets. Interestingly, finetuning a vision LLM using our VQA tasks reaffirms LLMs' bottleneck effect because the VQA tasks improve the LLMs' hierarchical consistency more than the vision LLMs'. We conjecture that one cannot make open-source vision LLMs understand visual concepts hierarchically until LLMs possess corresponding taxonomy knowledge. Code: `https://shorturl.at/sLZol`.

## 1 INTRODUCTION

Taxonomy is natural and core in visual understanding. The biology taxonomies cover many objects in our visual world (Van Horn et al., 2021); for example, a `Boston Terrier` belongs to the class of `Terrier`, which is a subtype of `Dog`, under `Mammal`, and ultimately part of the broader category `Animal`, forming a semantic path in the animal taxonomy: `Animal → Mammal → Dog → Terrier → Boston Terrier`. ImageNet (Deng et al., 2009) expands from the WordNet (Miller, 1995) taxonomy. Visual parts (Lee & Seung, 1999; Fidler & Leonardis, 2007; Arbeláez et al., 2012), attributes (Farhadi et al., 2009; Lampert et al., 2009; Palatucci et al., 2009), and relationships (Krishna et al., 2017) can be grouped hierarchically due to shared characteristics.

A high-performing, general-purpose visual understanding system should map visual inputs to both fine-grained leaf of a taxonomy and coarse-grained inner nodes. Meanwhile, it should label an input hierarchically consistently along the path that traces a leaf up to the root. Figure 1 illustrates a case selected from our experiments that the model predictions lack *hierarchical consistency*, failing to follow the path of `Animal → Vertebrate → Fish → Spiny-finned Fish → Anemone Fish`.

Surprisingly, little has been done to assess the hierarchical visual understanding performance of vision large language models (VLLMs) (Bai et al., 2025; Li et al., 2025; Chen et al., 2024; Zhu et al., 2025; Liu et al., 2023; Li et al., 2025), which have the potential to make such a general-purpose vision system. Indeed, VLLMs unify various vision tasks (e.g., visual recognition (Deng et al., 2009), captioning (Chen et al., 2015), question answering (Antol et al., 2015), and retrieval (Young et al., 2014)) into one model by anchoring visual encoders (Radford et al., 2021; Zhai et al., 2023; Cherti et al., 2023; Oquab et al., 2023) to a versatile pretrained LLM (Grattafiori et al., 2024; Yang et al., 2024), typically orders of magnitude bigger, offering integrated interactions with humans that involve images and videos in conjunction with natural language prompts. Comprehensively benchmarking VLLMs is essential for realizing their potential and identifying opportunities for improvements. Extensive benchmarks have recently emerged, such as the bilingual MMBench (Liu et al., 2024c), manually labeled MME (Fu et al., 2023), and MMMU (Yue et al., 2024) collected from college exams. We refer readers to (Zhang et al., 2024a) for an extensive list.

This work systematically evaluates VLLMs' hierarchical visual understanding capabilities using six taxonomies and four hierarchical image classification datasets. Conventionally, hierarchical image classification (Silla & Freitas, 2011; Park et al., 2025; Wu et al., 2024; Yi et al., 2022) aims to classify

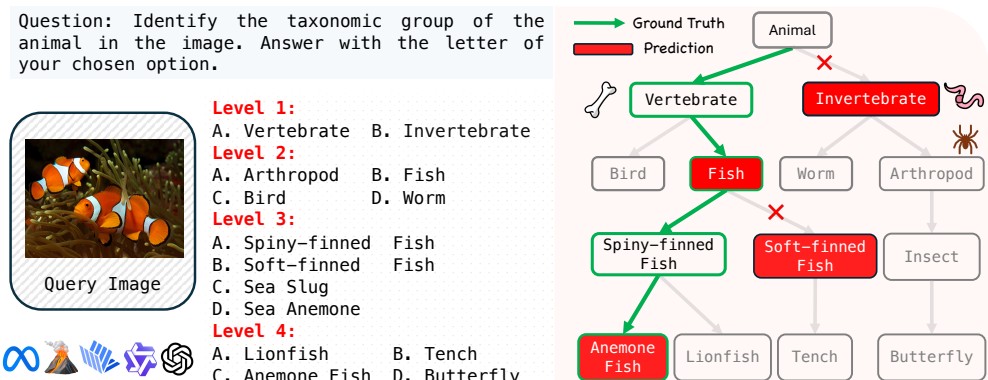

Figure 1: **Left**: Four-choice VQA tasks for evaluating VLLMs' hierarchical visual understanding. **Right**: A VLLM's answers (`in red boxes`) deviate from the ground truth path (**green arrows**), illustrating its lack of hierarchical consistency.

visual inputs into semantically structured categories across multiple levels of specificity, in contrast to flat classification, which treats labels as mutually exclusive and unstructured. We construct about one million four-choice visual question-answering (VQA) tasks from the hierarchical datasets (see Figure 1 for some examples). The tasks traverse all taxonomy levels, and the four choices of an individual task are from the same level. When evaluating VLLMs' performance over these tasks, we stress hierarchical consistency because it is unique to hierarchical visual understanding and crucial for adaptability to users' varying granularity preferences (Park et al., 2025; Conti et al., 2025; Wu et al., 2024).

Our main findings are as follows. First of all, many state-of-the-art VLLMs struggle with our VQA tasks, substantially lacking hierarchical consistency. For example, Qwen2.5-VL-72B (Bai et al., 2025) makes mistakes over 67% of the hierarchical paths in the iNaturalist (Van Horn et al., 2021) taxonomy. Moreover, in our attempt to tracing down the error causes, we find that LLMs are the bottleneck and lack taxonomy knowledge about the visual world — we believe this conclusion is true for open-source VLLMs, but we urge readers not to extrapolate it to proprietary LLMs because we could not probe their intermediate embeddings. In contrast, the visual encoder and projector modules demonstrate the ability to retain highly discriminative and well-structured visual features. We further show that the LLM embeddings about the visual concepts contain sufficient hierarchical cues and organize them orthogonally, but the model cannot decode them. Finally, finetuning a VLLM using our VQA tasks enhances its LLM's (text) hierarchical consistency more than the VLLM's (visual) hierarchical consistency, reaffirming LLMs' bottleneck effect to some extent.

## 2 VLLMs LACK HIERARCHICAL CONSISTENCY IN VISUAL UNDERSTANDING

We construct six hierarchical image classification benchmarks in a four-choice VQA format to systematically assess VLLMs' accuracy and hierarchical consistency in visual understanding. These benchmarks leverage datasets with inherent taxonomic structures, either derived from WordNet (Miller, 1995) or grounded in biological classification standards. In what follows, we formally define hierarchical image classification, followed by two evaluation metrics about accuracy and consistency, respectively. We then describe our VQA tasks and the first set of experiment results in this work.

### 2.1 HIERARCHICAL IMAGE CLASSIFICATION: NOTATIONS AND PROBLEM STATEMENT

General image classification tasks typically assume a flat label space, where each image $x \in \mathcal{X}$ is assigned a class label $y \in \mathcal{Y}$ out of a predefined set $\mathcal{Y}$ of mutually exclusive categories. However, many real-world problems exhibit rich semantic structures, in which labels are naturally organized into a hierarchy $\mathcal{T} = (\mathcal{Y}, \mathcal{E})$ (Park et al., 2025; Wu et al., 2024; Yi et al., 2022; Xia et al., 2023), such as a tree or a directed acyclic graph. Here, $\mathcal{E} \subseteq \mathcal{Y} \times \mathcal{Y}$ denotes the set of directed edges representing parent-child relationships, where $(y_i, y_j) \in \mathcal{E}$ indicates that $y_i$ is the parent of $y_j$ in the hierarchy. In hierarchical image classification, the objective is not only to predict the leaf node label $y \in \mathcal{Y}_{leaf} \subseteq \mathcal{Y}$ but also to correctly recover its full ancestral path $(y_0, y_1, \cdots, y_L)$ in $\mathcal{T}$, where $y_0$

denotes the root node and $L$ is the depth of the hierarchy. In this paper, we aim to evaluate VLLMs' hierarchical image classification capabilities, identify their limitations and underlying causes, and enhance their performance based on these insights.

## 2.2 Two Evaluation Metrics about Accuracy and Consistency, Respectively

For evaluation, we mainly focus on the hierarchical consistency of model predictions (Wu et al., 2024; Park et al., 2024b). Besides, we are interested in the leaf-level classification accuracy (Zhang et al., 2024b; Liu et al., 2024b; He et al., 2025), which can be viewed as the upper bound of the hierarchical consistency, detailed below.

**Hierarchical Consistent Accuracy (HCA)** (Wu et al., 2024). This metric is defined as

$$\text{HCA} = \frac{1}{N} \sum_{i=1}^{N} \prod_{j=1}^{L^i} \mathbb{1} \left[ f_\theta \left( x^i; \mathcal{Y}_j \right) = y_j^i \right],\tag{1}$$

where $N$ is the number of images in the testing set, $L^i$ denotes the depth of the hierarchy for the $i$-th input $x^i$ and may vary for different tasks in uneven trees, $f_\theta : \mathcal{X} \mapsto \mathcal{Y}$ is an image classifier, $\mathcal{Y}_j$ represents the set of labels at the $j$-th layer of the hierarchy, and $\mathbb{1}[\cdot]$ is an indicator function. HCA evaluates whether a model's predictions are consistent with the entire hierarchical path from the root to a leaf node. Specifically, it measures the proportion of samples for which all ancestor nodes along the predicted paths match the ground truth. This is a stricter metric than flat accuracy and serves as our primary evaluation criterion for hierarchical classification.

**Leaf-Level Accuracy** $\text{Acc}_{\text{leaf}}$ (Zhang et al., 2024b; Liu et al., 2024b; He et al., 2025). It cares about the predictions at the most fine-grained level of a taxonomy:

$$\text{Acc}_{\text{leaf}} = \frac{1}{N} \sum_{i=1}^{N} \mathbb{1} \left[ f_\theta \left( x^i; \mathcal{Y}_L \right) = y_L^i \right].\tag{2}$$

Interestingly, $\text{Acc}_{\text{leaf}}$ upper-bounds HCA because correctly assigning a leaf label $y_L$ to an input $x$ contributes to $\text{Acc}_{\text{leaf}}$, but correct leaf-level prediction does not increase HCA unless the model makes no mistake over all nodes in the path $(y_0, y_1, \cdots, y_L)$ connecting the leaf node to the root.

## 2.3 VQA Tasks Derived from Hierarchical Image Classification Datasets

VLLMs are the image classifiers $f_\theta$ in equations (1) and (2), and one can use language prompts to steer their output to a particular taxonomy level. More concretely, we formalize a VQA task for each image given a desired taxonomy level, $(x^i, \mathcal{Y}_j), i = 1, 2, \cdots, N, j = 1, 2, \cdots, L^i$, as follows.

**VQA Tasks.** We derive approximately one million four-choice VQA tasks and six taxonomies from four hierarchical image classification datasets (Wah et al., 2011; Van Horn et al., 2021; Deng et al., 2009; Bossard et al., 2014) to evaluate VLLMs in a closed-set setting. This setting mitigates the challenge of open-set generation, which involves a prohibitively large prediction space (Zhang et al., 2024b) and ambiguous prediction granularity. We test different VQA prompts (provided in Appendix C), and they generally follow this format:

```
<image> Given the plant in the image, what is its taxonomic classification
at the <hierarchy> (e.g., kingdom) level?
A.<similar class> B.<ground truth> C.<similar class> D.<similar class>
Answer with the option letter only.   (Choices are shuffled in the experiments)
```

Arguably, the four-choice VQA tasks are easier than the conventional hierarchical image classification, whose label space is orders of magnitude bigger than four. To compensate this difference, we make sure the four choices are from the same level of a taxonomy and use "confusing labels" in the VQA tasks. Specifically, we use SigLIP (Zhai et al., 2023) to compute the cosine similarity scores between an image and all text labels other than the ground truth (at a particular taxonomy level), selecting the top three most similar labels as the distracting VQA choices. Besides, we provide the results of randomly sampled choices in Appendix B.

**Hierarchical Image Classification Datasets.** CUB-200-2011 (CUB-200) (Wah et al., 2011) is a fine-grained bird dataset containing 200 species. We prompt GPT-4o (OpenAI, 2024) to map each

Table 1: The hierarchical consistent accuracy (HCA) and leaf-level accuracy $Acc_{leaf}$ of six open-source VLLMs, four CLIP-style models, and the proprietary GPT-4o.

| Model | iNat21-Animal | | iNat21-Plant | | ImgNet-Artifact | | ImgNet-Animal | | CUB-200 | |
|---|---|---|---|---|---|---|---|---|---|---|
| | HCA | $Acc_{leaf}$ | HCA | $Acc_{leaf}$ | HCA | $Acc_{leaf}$ | HCA | $Acc_{leaf}$ | HCA | $Acc_{leaf}$ |
| **Open-Source VLLMs** | | | | | | | | | | |
| LLaVA-OV-7B | 4.53 | 26.47 | 4.46 | 27.51 | 17.15 | 80.77 | 34.36 | 65.50 | 11.51 | 44.23 |
| InternVL2.5-8B | 8.52 | 27.65 | 5.56 | 28.36 | 21.42 | 78.07 | 37.82 | 65.19 | 22.07 | 45.56 |
| InternVL3-8B | 11.93 | 35.40 | 8.68 | 36.39 | 17.87 | 77.50 | 42.31 | 69.41 | 25.75 | 50.52 |
| Qwen2.5-VL-7B | 19.43 | 41.33 | 17.67 | 41.61 | 16.47 | 85.20 | 56.00 | 80.01 | 43.76 | 65.50 |
| Qwen2.5-VL-32B | 26.90 | 46.98 | 24.64 | 48.57 | 26.30 | 84.51 | 62.23 | 80.48 | 56.80 | 69.00 |
| Qwen2.5-VL-72B | 35.73 | 54.20 | 32.82 | 55.00 | 21.08 | 85.61 | 64.08 | 80.52 | 66.36 | 75.04 |
| **CLIP Models** | | | | | | | | | | |
| OpenCLIP | 1.04 | 23.53 | 0.19 | 28.12 | 9.11 | 83.64 | 12.57 | 81.14 | 4.31 | 80.39 |
| SigLIP | 2.15 | 12.71 | 0.46 | 18.84 | 6.41 | 87.19 | 24.40 | 86.85 | 23.18 | 73.84 |
| BioCLIP | 17.61 | 88.13 | 11.67 | 89.47 | 1.23 | 28.38 | 6.15 | 57.73 | 51.49 | 82.83 |
| BioCLIP2 | 41.84 | 95.94 | 37.91 | 95.26 | 1.38 | 55.47 | 8.34 | 72.57 | 55.80 | 92.94 |
| **Proprietary VLLM** | | | | | | | | | | |
| GPT-4o | 42.95 | 63.79 | 35.53 | 62.95 | 27.57 | 86.05 | 67.69 | 85.50 | 81.96 | 87.25 |

class to a four-level taxonomy: Order → Family → Genus → Specie. In addition, we incorporate the iNaturalist-2021 (iNat21) dataset (Van Horn et al., 2021), a large-scale collection with species-level annotations spanning various biological taxa. We separate it into two taxonomies, Plant and Animal, comprising 4,271 and 5,388 leaf nodes, respectively, and six levels. To increase structural diversity, we also experiment with ImageNet-1K (ImgNet) (Deng et al., 2009), whose leaf labels are coarser-grained than iNat21 and CUB-200. ImgNet is built upon the WordNet (Miller, 1995). We extract two relatively well-structured subsets from ImgNet: ImgNet-Animal and ImgNet-Artifact, following (Wu et al., 2024). We further refine these subsets to improve label quality and semantic consistency. Food-101 (Bossard et al., 2014) is about food classification, and its hierarchy is constructed based on the recent work of Liang & Davis (2025). More details are provided in Appenidx A.

## 2.4 EXPERIMENTS AND FINDINGS

We mainly study state-of-the-art open-source VLLMs: The Qwen2.5-VL (Bai et al., 2025) models of 7B, 32B, and 72B parameters, InternVL2.5-8B (Chen et al., 2024), InternVL3-8B (Zhu et al., 2025), and LLaVA-OV-7B (Li et al., 2025). Meanwhile, we include the proprietary GPT-4o's results for reference; in general, GPT-4o slightly outperforms Qwen-2.5-VL-72B, but the main findings below still apply. Besides, we experiment with two CLIP-style (Radford et al., 2021) general-purpose models, SigLIP-SO400M (Zhai et al., 2023) and OpenCLIP-L (Cherti et al., 2023), following the experiment protocol in (Radford et al., 2021) except that the candidate labels for each test image are restricted to the same four choices as fed to VLLMs. Finally, we evaluate two domain-specific CLIP-style models, BioCLIP (Stevens et al., 2024) and BioCLIP2 (Gu et al., 2025) under our experimental setup. Table 1 shows the results about the models' hierarchical consistency (HCA) and leaf-level accuracy ($Acc_{leaf}$) on iNat21, ImgNet, and CUB-200. The Food-101 results are in Appendix B to save space in the main text. We draw the following conclusions.

**VLLMs Lack Hierarchical Consistency in Visual Understanding.** Regardless of the leaf-level accuracy, all open-source VLLMs, CLIP models (both general and domain-specific), and GPT-4o lack hierarchical consistency because their HCA is significantly lower than $Acc_{leaf}$ (up to 99.3% relatively). The gaps on iNat21-Plant are especially big (e.g., 32.82 vs. 55.00 for Qwen2.5-VL-72B and 35.53 vs. 62.95 for GPT-4o). While one might expect better results on ImgNet, neither open-source VLLMs nor GPT-4o can make their HCA match $Acc_{leaf}$ — more than 20% decrease for all models, indicating that VLLMs make many mistakes along the paths from the taxonomies' roots to the leaf nodes even when they are correct over the leaves.

**Fine-Grained Visual Recognition Remains Challenging for VLLMs.** While the general-purpose VLLMs and CLIP models perform moderately well on ImgNet, they struggle with fine-grained object recognition; on the iNat21 dataset, even the best-performing GPT-4o gives rise to only 63% leaf-level

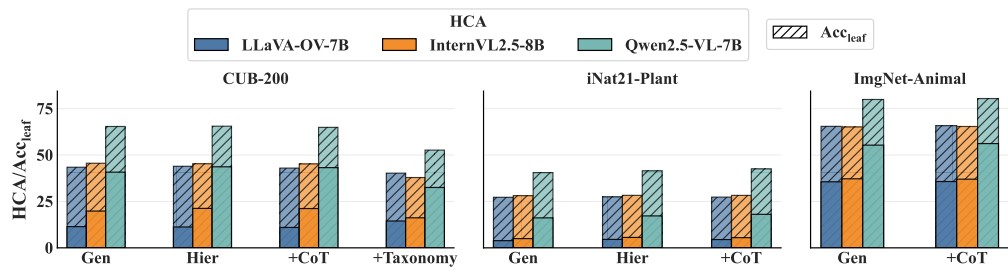

Figure 2: Prompt variants and their effects on VLLMs' hierarchical consistency (HCA) and fine-grained recognition $\text{Acc}_{\text{leaf}}$ (**Gen**: general prompts, **Hier**: hierarchical prompts, **+CoT**: prompts with Chain-of-Thought reasoning, **+Taxonomy**: prompts that include an explicit taxonomy in the JSON format. Please see Appendix C for details and examples.).

accuracy. Notably, InternVL2.5 and LLaVA-OV's results (about 27%) on iNat21 are only slightly above random guess (25%), and the general CLIP models are barely on par with random guess. In contrast, the domain specific model, BioCLIP2, yields 95.26% leaf-level accuracy on iNat21 and 92.94% accuracy on CUB-200, outperforming the general-purpose VLLMs. These findings are consistent with the recent work (Geigle et al., 2024; Zhang et al., 2024b; He et al., 2025; Yu et al., 2025) that recognizes the limitation of VLLMs on (fine-grained) image classification.

**Scaling Law Works for Hierarchical Visual Understanding.** Both hierarchical consistency and leaf-level accuracy improve as the size of the Qwen2.5-VL series of models increases. Moreover, the gap between HCA and $\text{Acc}_{\text{leaf}}$ progressively narrows. However, the largest models (Qwen2.5-VL-72B and GPT-4o) are still unsatisfactory in terms of both hierarchical consistency and fine-grained recognition, especially on the iNat21 benchmark.

**Qwen2.5-VLs Are Among the Most Powerful Open-Source VLLMs.** LLaVA-OV-7B's hierarchical consistency and leaf-level accuracy are below InternVLs and Qwen2.5-VLs. InternVL3-8B improves upon InternVL2.5-8B, but it is still under par with Qwen2.5-VL-7B.

**Domain-Specific BioCLIPs Lack Hierarchical Consistency.** While the domain-specific BioCLIPs yield superb leaf-level accuracy on the iNat21 taxonomies, they still lack hierarchical consistency; there exist persistent gaps between BioCLIPs' HCA and leaf-level accuracy on all the datasets. It is also worth noting that BioCLIPs' performance on ImgNets' leaf levels is below that of the general CLIP models, likely due to their (overly) specialization in the biological domain.

## 3 WHY ARE VLLMS POOR AT HIERARCHICAL IMAGE CLASSIFICATION?

We systematically investigate potential causes of VLLMs' low performance on hierarchical visual understanding. We first extensively study prompt variations in Section 3.1 and reveal that some prompts can lead to marginally better results than the rest, but the results remain generally bad. We then examine VLLMs' visual encoders and subsequent visual tokens to see whether and where essential visual information is lost when it forwards through VLLMs (Section 3.2). Interestingly, the discriminative cues in the visual tokens are maintained across various stages of the VLLM architectures, leading to about the same hierarchical image classification results immediately after the visual encoder, after the projection to the language token space, and at the very last layer of an LLM. *Finally and surprisingly, we find that the generally believed powerful LLMs, even the one with 72B parameters in our experiments, lack the taxonomy knowledge of our visual world,* hence likely responsible for VLLMs' poor performance on hierarchical visual understanding!

### 3.1 LANGUAGE PROMPTS ARE *Not* THE BOTTLENECK

Prompt engineering often comes as a remedy for boosting VLLMs' performance in different applications (Brown et al., 2020; Wang et al., 2024; Zhang et al., 2024b; Wu et al., 2024). Could it also rescue VLLMs on our hierarchical visual understanding tasks? We strive to test prompt variants comprehensively. We specify the taxonomy levels in the prompts for CUB-200 (Wah et al., 2011) and iNat21 (Van Horn et al., 2021), whose taxonomies are grounded in biology. We even add CUB-200's complete taxonomy as a JSON file to the prompts. For the other datasets with more

Table 2: (Text) HCA of VLLMs' LLMs and its correlation $\rho$ with VLLMs' (visual) HCA

| LLM of | iNat21-Animal | iNat21-Plant | ImgNet-Artifact | ImgNet-Animal | CUB-200 | $\rho$(text,visual) |
|---|---|---|---|---|---|---|
| LLaVA-OV-7B | 11.56 | 28.49 | 29.27 | 56.93 | 33.45 | 0.9116 |
| InternVL2.5-8B | 38.15 | 41.15 | 35.32 | 66.11 | 49.11 | 0.8832 |
| InternVL3-8B | 54.20 | 47.49 | 31.86 | 69.92 | 59.87 | 0.9030 |
| Qwen2.5-VL-7B | 52.08 | 64.21 | 35.06 | 68.14 | 63.86 | 0.8640 |
| GPT-4o | 96.85 | 96.70 | 42.31 | 89.56 | 98.81 | 0.7980 |

generic taxonomies, we test general and chain-of-thought (Kojima et al., 2022; Wei et al., 2022) prompts derived from the template in Section 2.3. Appendix C provides all prompts in detail, and Figure 2 shows the results of some high-performing prompts. We can see from the results that prompt design alone is insufficient to improve VLLMs' hierarchical consistency or leaf-level accuracy.

## 3.2 VISUAL EMBEDDINGS ARE *Not* THE BOTTLENECK

The open-source VLLMs in this work vary in specific implementations, but their core components are the same: A visual encoder mapping images to embeddings, a projector translating visual embeddings into the language token space, and an LLM. If the hierarchical structure and discriminativeness are lost before the visual embeddings reach LLMs, the overall VLLMs would inevitably perform poorly on our hierarchical visual understanding tasks. Hence, it is crucial to examine the visual embeddings. We train three linear classifiers per taxonomy level to respectively probe the visual encoder, projector, and last layer of an LLM, where the image representations are an average of the visual tokens. See Appendix C for further probing details and results.

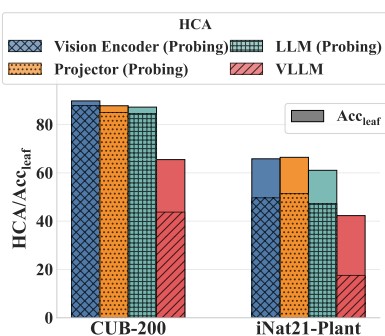

Figure 3: Qwen2.5-VL-7B vs. linearly probing the visual tokens at various stages of Qwen2.5-VL-7B on CUB-200 and iNat21-Plant.

Figure 3 shows the probing results of Qwen2.5-VL-7B over CUB-200 (Wah et al., 2011) and iNat21-Plant (Van Horn et al., 2021). Remarkably, the linear classifiers outperform Qwen2.5-VL-7B all around. They achieve not only higher leaf-level accuracy than Qwen2.5-VL but also much better hierarchical consistency, even though the classifiers of different taxonomy levels are independently trained. Moreover, the linear probing results remain about the same at different stages of the forward propagation (i.e., immediately after the visual encoder, projector, and last layer of the VLLM), indicating that the visual tokens remain discriminative and structurally rich throughout different LLM layers. These results are a strong defense for the visual embeddings: They carry sufficient hierarchical and discriminative cues and should not be blamed for VLLMs' poor hierarchical visual understanding performance.

## 3.3 *LLMs Are the Bottleneck* IN VLLMS' HIERARCHICAL VISUAL UNDERSTANDING

The huge discrepancy between the results of linearly probing visual tokens and VLLM performance in Figure 3 propels us to investigate other potential causes of VLLMs' low hierarchical consistency beyond the visual embeddings, and we find that the LLMs are the bottleneck.

### 3.3.1 OPEN-SOURCE VLLMS' LLMS LACK TAXONOMY KNOWLEDGE

We separate LLMs from open-source VLLMs and examine how much they know about the taxonomies used in our experiments. Mechanically, we reformulate our VQA tasks to a text-only version by replacing the images with their corresponding leaf labels:

```
Given the <leaf node label> (e.g., Anemone Fish), what is its taxonomic
classification at the <hierarchy> (e.g., kingdom) level?
A.<similar class>  B.<ground truth>  C.<similar class>  D.<similar class>
Answer with the option letter only.  (Choices are shuffled in the experiments)
```

This process produces 0.7 million QA tasks after deduplication. We use them to assess LLMs and report the (text) HCA results in Table 2 — we use (text/visual) HCA to refer to LLMs/VLLMs'

performance on text/visual QA tasks. We find that Qwen2.5-VL's LLM achieves only 63.86% (text) HCA on CUB-200, whose taxonomy comprises merely four levels. The LLMs of LLaVA-OV and InternVL-2.5 give rise to even lower (text) HCAs on CUB-200 (33% and 49%).

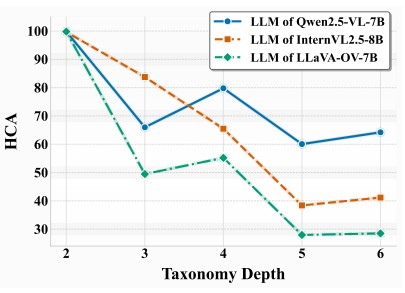

Figure 4: Text HCA of different VLLMs' LLMs over the iNat21-Plant taxonomies of various depths.

One might wonder if those low (text) HCAs are due to that the biology taxonomy underlying CUB-200 is too specific for general LLMs. However, Table 2 further reveals that the LLMs also cannot perform well on ImgNet's general taxonomies. Besides, we progressively simplify our QA tasks by chopping the iNat21-Plant taxonomy level by level. Figure 4 plots the (text) HCA results, which increase as the taxonomy becomes shallower (and, correspondingly, the leaf nodes are less fine-grained). Still, they are below 90% regardless of the taxonomies' depths. There are noticeable drops at Levels 3 and 5 for Qwen2.5-VL and LLaVA-OV's LLMs, implying that they pose more challenges than the other levels for the LLMs' hierarchical reasoning. *These results are highly unexpected*, given the recent success of LLMs over various benchmarks and domains (OpenAI, 2024; Team et al., 2023; Yang et al., 2024; Liu et al., 2024a; Yang et al., 2025).

**Correlation between (text) HCA and $Acc_{leaf}$-scaled (visual) HCA.** An LLM's low (text) HCA undoubtedly discounts its corresponding VLLM's hierarchical consistency on visual inputs. We can quantify this notion using Pearson's correlation coefficient. Since the (text) HCA's corresponding leaf-level accuracy is 100% — we replaced images with their ground-truth leaf labels when making the text QA tasks, we normalize (visual) HCA by $1/Acc_{leaf}$. The last column in Table 2 shows that the correlation between (text) HCA and $Acc_{leaf}$-scaled (visual) HCA is as high as 0.9116.

*A note about GPT-4o's (text) HCA.* The analyses above apply to only open-source VLLMs because we cannot separate LLMs from the proprietary GPT-4o. Unlike the open-source LLMs' low (text) HCA, GPT-4o's (text) HCA scores are as high as 98.81. Hence, we conjecture that the LLM part is likely not GPT-4o's bottleneck in hierarchical visual understanding; instead, there are other possible causes of GPT-4o's hierarchical inconsistency about the visual world.

### 3.3.2 WHY ARE LLMs POOR AT HIERARCHICAL *Text* CLASSIFICATION?

In what follows, we present some preliminary quests into why and where LLMs fail at the seemingly simple hierarchical four-choice text classification tasks. We rule out the vision-language tuning that anchors visual encoders to pretrained LLMs and conclude that the language decoders are responsible for LLMs' lack of taxonomy knowledge.

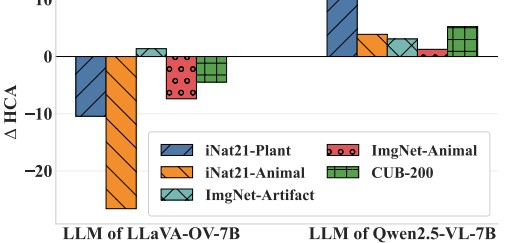 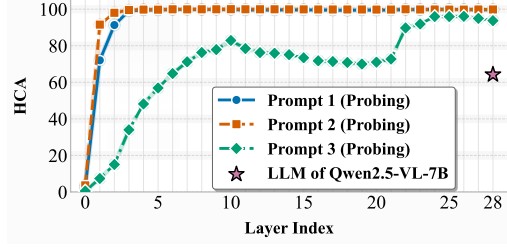

Figure 5: **Left**: (Text) HCA difference between vision-language-tuned LLMs and the original ones. **Right**: (Text) HCA of linearly probing different layers of Qwen-2.5-VL-7B's LLM on iNat21-Plant.

**Vision-Language Tuning Is *Not* the Reason.** Acute readers likely have noted that our previous LLM results are about the LLM parts of VLLMs, not the "true" standalone LLMs. Does the vision-language tuning, which is needed when one connects a visual encoder with an LLM, compromise LLMs and potentially induce catastrophic forgetting of taxonomy knowledge?

We answer this question by studying the original LLMs from which VLLMs are initialized, using the same text-only hierarchical classification setup described in Section 3.3. Figure 5 (Left) compares LLaVA-OV-7B and Qwen2.5-VL-7B's LLMs with their corresponding original LLMs. First of all, we see that the original LLMs are on par with or even worse than their vision-tuned counterparts,

indicating that the standalone LLMs still lack a strong grasp of taxonomy knowledge. Interestingly, Qwen2.5-VL's LLM actually outperforms its original LLM on all taxonomies; in other words, the vision-language tuning actually enhances the LLM's (text) hierarchical consistency. In contrast, LLaVA-OV's vision-language tuning weakens the LLM's (text) HCA.

**LLMs Encode Hierarchical Structures Effectively but Cannot Decode Them Sufficiently.** Next, we shift attention to the LLM embeddings of the concepts in our taxonomies — if the embeddings do not provide sufficient hierarchical structural cues, there is little chance LLMs can decode them. To this end, we convert a taxonomy into language prompts of three variants:

> **Prompt 1:** `<leaf node label>` (e.g., Blue Jay) belongs to the `<hierarchy>` (e.g., Order) of `<ground truth>` (e.g., Passeriformes).
> **Prompt 2:** Given the `<leaf node label>`, what is its taxonomic classification at the `<hierarchy>` level? It belongs to `<ground truth>`.
> **Prompt 3:** Given the `<leaf node label>`, what is its taxonomic classification at the `<hierarchy>` level?

We then train a linear classifier for each taxonomy level to probe the average embedding of the language tokens in every layer of an LLM. Figure 5 (Right) summarizes the (text) HCA results of Qwen2.5-VL-7B's LLM on iNat21-Plant: The text embeddings give rise to highly hierarchically consistent linear probes. Even for Prompt 3, with the ground-truth hierarchy labels withheld, the linear probes that receive only the leaf node embeddings can still achieve near-perfect hierarchical consistency in the LLM's deeper layers. In other words, the specialized linear probes can decode the taxonomy knowledge significantly better than the general-purpose LLM.

**LLMs' Hierarchical Orthogonality Does Not Guarantee Hierarchical Consistency.** Park et al. (2024a) recently predicted that LLMs represent hierarchical relations orthogonally in the representation space, e.g., `animal` is orthogonal to `bird−mammal`. They validated the prediction using Gemma (Team et al., 2024) and LLaMA (Grattafiori et al., 2024), and we further verify it in Figure 6 using both the original Qwen2.5-7B and the one after vision-language tuning. This pleasant geometric interpretation is, unfortunately, shadowed by the poor performance of Gemma and Qwen2.5-7B on our taxonomy QA tasks — we report the Gemma results in Appendix C. We argue that more fine-grained analyses of the LLM representation are required to establish a relationship between LLMs' hierarchical consistency and geometry.

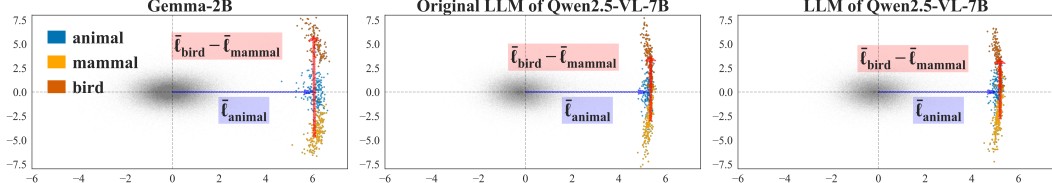

Figure 6: Hierarchical semantics are encoded as orthogonality in different LLMs' representation spaces (figures drawn following (Park et al., 2024a)).

## 4    LLMS GAIN MORE CONSISTENCY THAN VLLMS FROM FINETUNING

Could we improve the VLLMs' hierarchical visual understanding capabilities via finetuning using our VQA tasks built upon taxonomies? Likely, no, because LLMs are the bottleneck: The LLMs' hierarchical consistency over text-only tasks is so bad (Table 2) that we conjecture this shortcoming can only be fixed in the pretraining stage rather than the "tail patching" finetuning stage.

Still, the following presents some LoRA-finetuning (Hu et al., 2022) experiments with Qwen2.5-VL-7B, the best-performing 7B VLLM in our previous experiments, mainly for two reasons. One is to see how much finetuning could help, even though we believe pretraining instead of finetuning should be the rescue to VLLMs' hierarchical inconsistency. The other is further to investigate the interplay between VLLMs and their LLMs — interestingly, our results reaffirm that LLMs are the bottleneck for VLLMs' hierarchical visual understanding because LLMs' performance gain from the finetuning upper-bounds VLLMs'. Our finetuning data consists of VQA tasks constructed from iNat21-Plant's training set. We then evaluate the finetuned model's improvement on iNat21-Plant, its generalization to other hierarchical visual understanding datasets, and how well it maintains the general vision-language capabilities. Please see Appendix D for more details on the training.

Table 3: (Visual) HCA and $Acc_{leaf}$ of Qwen2.5-VL-7B before and after the LoRA-finetuning.

| Model | iNat21-Animal | | iNat21-Plant | | ImgNet-Animal | | CUB-200 | |
|---|---|---|---|---|---|---|---|---|
| | HCA | $Acc_{leaf}$ | HCA | $Acc_{leaf}$ | HCA | $Acc_{leaf}$ | HCA | $Acc_{leaf}$ |
| Qwen2.5-VL-7B | 19.43 | 41.33 | 17.67 | 41.61 | 56.00 | 80.01 | 43.76 | 65.50 |
| Qwen2.5-VL-7B (LoRA) | 23.38 | 45.00 | 29.34 | 47.66 | 58.62 | 80.28 | 46.17 | 67.12 |
| $\Delta$ | +3.95 | +3.67 | +11.67 | +6.05 | +2.62 | +0.27 | +2.41 | +1.62 |

Table 4: (Text) HCA of the LLM of Qwen2.5-VL-7B before and after the LoRA-finetuning.

| Model | iNat21-Animal | iNat21-Plant | ImgNet-Animal | CUB-200 |
|---|---|---|---|---|
| LLM of Qwen2.5-VL-7B | 52.08 | 64.21 | 68.14 | 63.86 |
| LLM of Qwen2.5-VL-7B (LoRA) | 65.63 | 84.87 | 72.39 | 66.15 |
| $\Delta$ | +13.55 | +20.66 | +4.25 | +2.29 |

**Results and Discussion.** Tables 3 shows that finetuning Qwen2.5-VL using the VQA tasks that partially cover the iNat21-Plant taxonomy delivers improvements on both iNat21-Plant and other datasets. On iNat21-Plant, HCA rises from 17.67 to 29.34 (+11.67 absolute gain), while $Acc_{leaf}$ gains 6.05. The HCA on ImgNet-Animal increases from 56.00 to 58.62 and on CUB-200 from 43.76 to 46.17. More interestingly, Table 4 indicates that the LLM's (text) HCA increases more from the finetuning than Qwen2.5-VL's (visual) HCA (e.g., 20.66 vs. 11.67 on iNat21-Plant and 4.25 vs. 2.62 on ImgNet-Animal). To some extent, this finding reaffirms that LLMs are the bottleneck of VLLMs' hierarchical visual understanding, and one has to improve LLMs' (text) taxonomy knowledge to boost VLLMs' (visual) hierarchical consistency. Besides, our results demonstrate that vision-language training can benefit both VLLMs and their LLMs, aligning with some recent advocates for improving LLMs using multimodal data beyond language only (Li et al., 2023b; Tu et al., 2024). Appendix D reports more results, including that the finetuned model does not lose its general capability tested on MME (Fu et al., 2023), MMBench (Liu et al., 2024c), and SEED-Bench (Li et al., 2023a).

## 5 RELATED WORK

Hierarchical classification (Silla & Freitas, 2011; Kosmopoulos et al., 2015) is vital for a comprehensive understanding of the visual world (Yi et al., 2022; Park et al., 2024b; Zeng et al., 2024; Sinha et al., 2024; Chen et al., 2022; Park et al., 2025) and many language concepts (Zhou et al., 2020; Wang et al., 2022; Zhou et al., 2025; He et al., 2024; Nikishina et al., 2023; Lin & Ng, 2022). Several recent studies have revisited this longstanding problem and shown that CLIP-style (Radford et al., 2021) models lack consistency across taxonomic levels (Wu et al., 2024; Geng et al., 2023; Xia et al., 2023; Pal et al., 2024; Desai et al., 2023). Zhang et al. (2024b) first identified the limitations of current VLLMs in fine-grained image classification. Building on this, Liu et al. (2024b) further assess a broader range of VLLMs. Beyond closed-set evaluation (Yu et al., 2025; Geigle et al., 2024; He et al., 2025), Conti et al. (2025) benchmark VLLMs' open-world classification, while Snæbjarnarson et al. (2025) propose to evaluate VLLMs' open-set predictions using a taxonomic similarity rather than exact string matching. However, to the best of our knowledge, no prior work has examined VLLMs under the hierarchical visual understanding context. Please see Appendix E for details.

## 6 CONCLUSION

This work presents a systematic evaluation of state-of-the-art VLLMs's hierarchical visual understanding performance. We find that both open-source VLLMs and the proprietary GPT-4o give rise to low hierarchical consistency over six taxonomies of visual concepts. Probing results reveal that the visual and text embeddings carry rich hierarchical and discriminative cues, whereas the LLMs fail to decode them, implying LLMs are the bottleneck. Finetuning on hierarchical VQA tasks improves VLLMs' hierarchical consistency on visual inputs while preserving their performance on general VQA tasks. Intriguingly, the finetuning benefits the LLM's (text) hierarchical consistency more than the corresponding VLLM's (visual) hierarchical measure. Ingesting the taxonomy-knowledge gap to LLMs, likely during pretraining rather than post-hoc patching, is a promising path toward VLLMs that reason coherently across different levels of semantic granularity about the visual world.

## REPRODUCIBILITY STATEMENT

We are committed to reproducible research. This link, `https://shorturl.at/sLZol`, points to the anonymous, downloadable source code and data used in this work. A detailed README file is available with instructions for setup and a list of all necessary dependencies. Moreover, the appendix provides a comprehensive description of the data processing steps, LLM prompts, and experiment setups, ensuring that our results are replicable. Finally, we will set up a github.com repository after the paper review process to facilitate discussions of any reproducibility issues.

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

In the **appendices**, we provide all implementation details to promote the reproducibility of our work, more experimental results, and further discussions. Section A is about the hierarchical image and text classification datasets. Section B supplements the main experiments in the paper, including the models, experiment setups, quantitative and qualitative results, and ablation studies. Section C presents various prompting and linear probing results, including those with the Gemma model and larger VLLMs up to 72B parameters. Section D allows one to reproduce our finetuning results and shows comparison results of text-only finetuning and on general VQA benchmarks. Finally, Sections E, F, and G broaden the discussions by including more related works, limitations, and broader impacts of the work. Section H is the LLM usage statement.

# A    CURATION OF THE HIERARCHICAL CLASSIFICATION BENCHMARKS

## A.1    HIERARCHICAL IMAGE CLASSIFICATION BENCHMARKS

Following prior work on hierarchical image classification, we adopted several commonly used hierarchical classification datasets, including ImageNet (Deng et al., 2009), iNaturalist-2021 (Van Horn et al., 2021), CUB-200-2011 (Wah et al., 2011) and Food-101 (Bossard et al., 2014). Table 5 summarizes the six taxonomies and four datasets we use to construct the VQA tasks.

Table 5: Overview of the six taxonomies and four datasets we use to construct the VQA tasks.

| Dataset | #Levels | #Leaf Nodes | #Images | Hierarchy Distribution |
|---|---|---|---|---|
| CUB-200 (Wah et al., 2011) | 4 | 200 | 5,794 | 13-37-124-200 |
| iNat21-Plant (Van Horn et al., 2021) | 6 | 4,271 | 42.71K | 5-14-85-286-1702-4271 |
| iNat21-Animal (Van Horn et al., 2021) | 6 | 5,388 | 53.88K | 6-27-152-715-2988-5388 |
| ImgNet-Animal (Deng et al., 2009) | 11 | 397 | 19.85K | 2-10-37-81-123-81-65-41-64-34-2 |
| ImgNet-Artifact (Deng et al., 2009) | 7 | 491 | 24.55K | 5-40-147-204-162-62-44 |
| Food-101 (Bossard et al., 2014) | 4 | 84 | 21.00K | 6-29-40-24 |

Due to the inherent unconstrained nature of open-ended predictions by VLLMs, even when provided with detailed instructions, their performance in open-ended hierarchical classification remains extremely limited, with an $\mathrm{Acc_{leaf}}$ as low as 3.88% by Qwen2.5-VL-7B. To more effectively evaluate model performance, we construct approximately one million multiple-choice questions in a four-choice VQA format. We provide the data construction process of hierarchy VQA benchmarks shown in Figure 7. To better illustrate the data format, we also provide several examples from different datasets as shown in Figure 8.

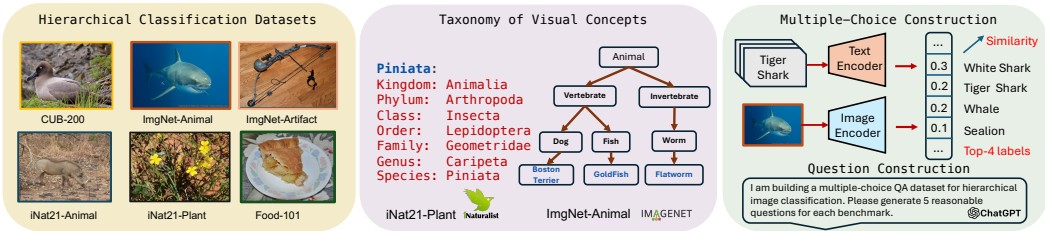

Figure 7: **Overview of hierarchical image classification benchmarks construction process.** Our hierarchical VQA benchmark is built on four datasets and covers six taxonomies. We first obtain the hierarchical structure for each taxonomy (biology standard and WordNet (Miller, 1995) semantics). Then, we use SigLIP (Zhai et al., 2023) to generate four choices for each image based on the image-text similarities, comprising the groundtruth class name and the top three classes returned by SigLIP. Finally, we leverage GPT-4o to generate the corresponding questions.

## A.2    TAXONOMY REFINEMENT

To refine the taxnonomy quality, we use Wikipedia and GPT-4o to carefully examine the hierarchical relations in each taxonomy.

System Prompt: "You are an expert in hierarchical image classification. Given an image, classify it at its current hierarchy level by selecting the most appropriate option from the provided choices (labeled with letters). Respond with only the corresponding letter."

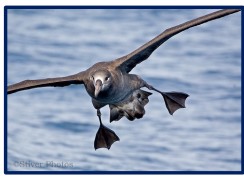

Based on the image, what is the taxonomic classification at the order level?
A. Anseriforme
B. Pelecaniforme
C. Procellariiformes
D. Podicipediformes
Answer with the option's letter from the given choices directly.

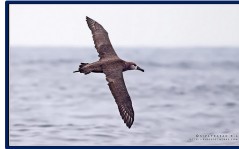

How can the bird in this image be categorized taxonomically?
A. Pomarine jaeger
B. Black-footed albatross
C. Laysan albatross
D. Sooty albatross
Answer with the option's letter from the given choices directly.

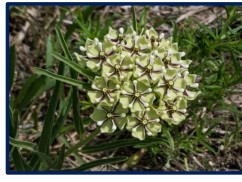

Given the plant in the image, what is its taxonomic classification at the order level?
A. Gentianales
B. Apiales
C. Cornales
D. Dipsacales
Answer with the option's letter from the given choices directly.

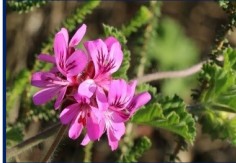

What is the systematic position of the plant in the image in the biological classification hierarchy?
A. Bailey
B. Draba
C. Erysimum
D. Barbarea
Answer with the option's letter from the given choices directly.

Figure 8: Examples of the prompt formats used in our four-choice hierarchical VQA tasks.

*Re-Verify the Taxonomies*: For example, in the original WordNet hierarchy, "indigo bunting" is misclassified under "finch". However, based on the established taxonomy, it belongs to the cardinal family. We corrected its hierarchical path to: animal → vertebrate → bird → oscine → cardinal → indigo bunting. We detected these errors using GPT-4o and then validated them using reliable taxonomies in iNaturalist and Wikipedia.

*Remove Ambiguous Paths*: For example, in WordNet, "tusker" is assigned to the overly coarse path: animal → vertebrate → mammal → tusker. However, "tusker" is merely a colloquial term for an elephant, and WordNet already includes a more fine-grained and taxonomically accurate path for "elephant": animal → vertebrate → mammal → placental → elephant → African elephant. We removed the "tusker" node, as it lacks specificity and overlaps with the existing, more precise elephant class.

*Correct Subordinate Relationships Within the Same Hierarchy Level*: In the ImgNet-Artifact dataset, the hierarchy provided by WordNet exhibits notable semantic inconsistencies, particularly where concepts at the same hierarchical level implicitly reflect subordinate relationships. For instance, under the category "device", concepts such as "machine", "instrument", "musical instrument", and "mechanism" are listed as siblings. However, "musical instrument" is a subtype of "instrument", making the latter a hypernym of the former. Treating these as peers can lead to ambiguous or conflicting answers when classifying an image, as both may be considered valid even though one is semantically nested within the other. To resolve these issues, we used GPT-4o to analyze whether sibling category pairs exhibited valid hypernym-hyponym relationships systematically. We refined or removed problematic intermediate nodes and eliminated leaf nodes associated with overly coarse or semantically inconsistent hierarchy paths.

### A.3 Hierarchical Text Classification Benchmarks

For each curated hierarchical image classification benchmark, we derive a text-only variant. Concretely, we replace the image token in each prompt with the leaf node label of the corresponding hierarchy, while preserving the original answer choices, which were deliberately selected as *confusing labels*. An example of the resulting prompt template is illustrated in Figure 9.

Figure 9: An example of the text QA construction from the hierarchical VQA.

## B Detailed Experiment Setup and Results

### B.1 Models

An overview of the models used in the evaluation experiments is provided in Table 6.

Table 6: Models used in evaluation experiments and their sources.

| Model | Source |
|---|---|
| LLaVA-OV-7B (Li et al., 2025) | https://huggingface.co/lmms-lab/llava-onevision-qwen2-7b-ov |
| InternVL2.5-8B (Chen et al., 2024) | https://huggingface.co/OpenGVLab/InternVL2_5-8B |
| InternVL3-8B (Zhu et al., 2025) | https://huggingface.co/OpenGVLab/InternVL3-8B |
| Qwen2.5-VL-7B (Bai et al., 2025) | https://huggingface.co/Qwen/Qwen2.5-VL-7B-Instruct |
| Qwen2.5-VL-32B (Bai et al., 2025) | https://huggingface.co/Qwen/Qwen2.5-VL-32B-Instruct |
| Qwen2.5-VL-72B (Bai et al., 2025) | https://huggingface.co/Qwen/Qwen2.5-VL-72B-Instruct |
| GPT-4o[1] (OpenAI, 2024) | https://openai.com/api/ |
| OpenCLIP (Cherti et al., 2023) | https://huggingface.co/laion/CLIP-ViT-L-14-laion2B-s32B-b82K |
| SigLIP (Zhai et al., 2023) | https://huggingface.co/google/siglip-so400m-patch14-384 |
| BioCLIP (Stevens et al., 2024) | https://huggingface.co/imageomics/bioclip |
| BioCLIP2 (Gu et al., 2025) | https://huggingface.co/imageomics/bioclip-2 |

### B.2 Hierarchical Evaluation Metrics

In addition to the metrics introduced in Section 2.2, we report results on three complementary metrics that probe different aspects of hierarchical classification ability.

**Point-Overlap Ratio (POR) (Yi et al., 2022).** To provide a more comprehensive evaluation of model performance across the full hierarchy, Yi et al. (2022) proposed the point-overlap ratio, defined as:

$$\text{POR} = \frac{1}{N} \sum_{i=1}^{N} \frac{\sum_{j=1}^{L_i} \mathbb{1}\left[ f_\theta\left(x_i; \mathcal{Y}_j\right) = y_j^i \right]}{L_i}. \tag{3}$$

Unlike HCA, which requires an exact match along the entire path, POR allows for partial correctness by computing the average proportion of correctly predicted nodes. This metric offers a more fine-grained view of model performance over the taxonomy and captures the extent to which predictions align with the target hierarchy.

**Strict Point-Overlap Ratio (S-POR).** S-POR sharpens the original POR criterion by rewarding only *contiguous* stretches of correct predictions. For the $i$-th sample, we locate the longest run of

---

[1] GPT-4o results reported in this paper use the `gpt-4o-2024-04-01-preview` model for image-based tasks and the `gpt-4o-2024-11-20` model for text-only evaluations.

consecutive correctly labelled layers and normalise by the hierarchy depth $L_i$:

$$\text{S-POR} = \frac{1}{N} \sum_{i=1}^{N} \frac{1}{L_i} \max_{1 \leq a \leq b \leq L_i} \left[ (b - a + 1) \prod_{j=a}^{b} \mathbb{1}\left[ f_\theta(x_i; \mathcal{Y}_j) = y_j^i \right] \right].$$

This stricter definition penalizes sporadic correctness and encourages full-path consistency.

**Top Overlap Ratio (TOR).** Following Wu et al. (2024), TOR measures *local* consistency by treating each pair of adjacent layers as an evaluation unit:

$$\text{TOR} = \frac{1}{N} \sum_{i=1}^{N} \frac{1}{L_i - 1} \sum_{j=1}^{L_i - 1} \mathbb{1}\left[ f_\theta(x_i; \mathcal{Y}_j) = y_j^i \right] \mathbb{1}\left[ f_\theta(x_i; \mathcal{Y}_{j+1}) = y_{j+1}^i \right].$$

A TOR value of 1 indicates that every neighbouring pair is correctly predicted, whereas lower scores reflect violations of pairwise hierarchical coherence.

### B.3    EVALUATION RESULTS WITH ALL METRICS

Table 7: Evaluation results across all VLMs on CUB-200, ImgNet-Animal, ImgNet-Artifact and iNat21-Plant with POR, S-POR and TOR reported.

| Model | CUB-200 | | | ImgNet-Animal | | | ImgNet-Artifact | | | iNat21-Plant | | |
|---|---|---|---|---|---|---|---|---|---|---|---|---|
| | POR | S-POR | TOR | POR | S-POR | TOR | POR | S-POR | TOR | POR | S-POR | TOR |
| **Open-Source VLLMs** | | | | | | | | | | | | |
| LLaVA-OV-7B | 58.46 | 42.06 | 35.01 | 83.56 | 70.56 | 72.36 | 63.36 | 26.33 | 44.74 | 55.50 | 43.08 | 37.09 |
| InternVL2.5-8B | 66.58 | 55.10 | 47.34 | 84.59 | 76.08 | 74.71 | 65.65 | 35.35 | 45.96 | 57.82 | 43.20 | 39.48 |
| InternVL3-8B | 69.80 | 53.62 | 51.75 | 86.34 | 79.33 | 77.72 | 62.96 | 31.35 | 42.48 | 62.54 | 48.83 | 44.72 |
| Qwen2.5-VL-7B | 80.85 | 70.52 | 67.97 | 90.52 | 84.52 | 83.84 | 64.12 | 26.47 | 44.53 | 71.95 | 59.37 | 57.45 |
| Qwen2.5-VL-32B | 86.86 | 81.62 | 78.71 | 92.14 | 87.89 | 87.00 | 69.25 | 38.82 | 50.55 | 76.14 | 65.37 | 63.90 |
| Qwen2.5-VL-72B | 89.79 | 86.20 | 83.63 | 92.43 | 88.74 | 87.77 | 67.21 | 30.86 | 48.10 | 80.23 | 70.92 | 69.86 |
| **CLIP Models** | | | | | | | | | | | | |
| OpenCLIP | 47.22 | 19.04 | 17.28 | 71.68 | 37.71 | 47.88 | 53.80 | 20.60 | 28.35 | 34.40 | 14.17 | 11.38 |
| SigLIP | 66.56 | 45.84 | 41.79 | 78.95 | 48.46 | 59.24 | 50.90 | 16.15 | 24.77 | 34.67 | 16.97 | 15.00 |
| BioCLIP | 83.99 | 65.63 | 71.82 | 55.36 | 25.01 | 28.29 | 29.03 | 9.13 | 8.73 | 69.80 | 31.13 | 47.62 |
| BioCLIP2 | 86.27 | 64.31 | 75.52 | 64.16 | 31.28 | 39.75 | 34.03 | 7.57 | 11.12 | 84.34 | 54.86 | 68.64 |
| **Proprietary VLLMs** | | | | | | | | | | | | |
| GPT-4o | 94.46 | 92.29 | 91.00 | 93.33 | 89.16 | 88.83 | 70.45 | 40.42 | 51.47 | 79.92 | 69.37 | 68.62 |

We report more comprehensive evaluation results in Table 7 and Table 8. From these tables, we observe that VLLMs achieve relatively high POR scores, indicating strong classification performance across different levels of granularity. However, both S-POR and TOR scores remain relatively low, reflecting inconsistency in the model predictions.

As the capacity of the VLLM increases (e.g., from Qwen2.5-VL 7B to 32B and 72B), the gap between POR and S-POR narrows, suggesting improved consistency in preserving the hierarchical structure during prediction. For GPT-4o, the gap between POR and S-POR on CUB-200 is only 2.17%, indicating that the correctly predicted nodes are mostly concentrated in the upper levels of the hierarchy. Additionally, the gap between TOR and POR also shrinks as model capacity increases, suggesting that better local hierarchical consistency is achieved.

While many individual nodes along the taxonomy path are predicted correctly, as evidenced by high POR scores, the probability of correctly predicting the entire path from root to leaf remains low. Although prior work (Conti et al., 2025) has noted that models often succeed in predicting coarse-grained categories but fail at fine-grained distinctions, our evaluation reveals that models sometimes predict the correct fine-grained label while misclassifying the corresponding coarse category. Therefore, beyond assessing fine-grained classification accuracy, it is equally important to evaluate the hierarchical consistency of VLLMs across different levels of granularity.

Table 8: Evaluation results across all VLMs on iNat21-Animal with all metrics reported.

| Model | $\text{Acc}_{\text{leaf}}$ | HCA | POR | S-POR | TOR |
|---|---|---|---|---|---|
| **Open-source VLLMs** | | | | | |
| LLaVA-OV-7B | 26.47 | 4.53 | 60.31 | 45.96 | 45.53 |
| InternVL2.5-8B | 27.65 | 8.52 | 66.26 | 57.07 | 53.50 |
| InternVL3-8B | 35.40 | 11.93 | 69.00 | 59.13 | 55.55 |
| Qwen2.5-VL-7B | 41.66 | 19.73 | 74.80 | 66.92 | 63.71 |
| Qwen2.5-VL-32B | 46.98 | 26.90 | 78.38 | 72.09 | 68.93 |
| Qwen2.5-VL-72B | 54.20 | 35.73 | 81.76 | 76.05 | 73.55 |
| **CLIP Models** | | | | | |
| OpenCLIP | 23.53 | 1.04 | 41.11 | 19.02 | 21.12 |
| SigLIP | 12.71 | 2.15 | 38.24 | 38.24 | 33.95 |
| BioCLIP | 88.13 | 17.61 | 72.46 | 29.10 | 55.77 |
| BioCLIP2 | 95.94 | 41.84 | 85.89 | 56.13 | 73.34 |
| **Proprietary VLLM** | | | | | |
| GPT-4o | 63.79 | 42.95 | 84.25 | 77.74 | 76.15 |

Compared with results in Table 1, models with higher POR, S POR, and TOR scores tend to exhibit better hierarchical consistency.

### B.4 ILLUSTRATIVE MISTAKES MADE BY VLLMS

We visualize some hierarchical prediction errors made by open-source VLLMs in Figure 10.

### B.5 RESULTS ON CUB-200 AND INAT21-PLANT WITH RANDOM CHOICES

Table 9: Hierarchical evaluation results (image) on CUB-200 and iNat21-Plant benchmarks with random choices.

| Model | HCA | $\text{Acc}_{\text{leaf}}$ | POR | S-POR | TOR |
|---|---|---|---|---|---|
| **CUB-200** | | | | | |
| LLaVA-OV-7B | 40.25 | 86.14 | 78.12 | 59.30 | 58.94 |
| InternVL2.5-8B | 64.20 | 91.06 | 88.36 | 77.13 | 76.91 |
| InternVL3-8B | 67.50 | 93.80 | 90.55 | 75.96 | 82.23 |
| Qwen2.5-VL-7B | 82.34 | 97.15 | 95.05 | 87.78 | 90.23 |
| **iNat21-Plant** | | | | | |
| LLaVA-OV-7B | 28.41 | 69.04 | 75.36 | 58.53 | 60.03 |
| InternVL2.5-8B | 36.45 | 75.97 | 80.25 | 60.81 | 67.22 |
| InternVL3-8B | 51.94 | 89.70 | 87.19 | 70.96 | 76.28 |
| Qwen2.5-VL-7B | 70.09 | 93.76 | 92.75 | 82.88 | 86.15 |

As shown in Table 9, using random choices significantly improves the model's fine-grained accuracy-reaching up to 90% for Qwen2.5-VL. However, even with random choices, the gap between $\text{Acc}_{\text{leaf}}$ and HCA still exceeds 20%. For models like LLaVA-OV-7B and InternVLs, this gap is even more pronounced, reaching up to 40% on the iNat21-Plant benchmark, despite their relatively high $\text{Acc}_{\text{leaf}}$. Therefore, our conclusion and analysis are still valid regardless of how the choices are constructed. However, the random choice construction does not reflect real-world scenarios, as it drastically reduces the task difficulty: three out of the four choices are likely to be completely unrelated to the query concept. For VLLMs, constructing similar choices based on image-text similarity better reflects practical scenarios, as end users are more likely to compare closely related concepts rather than unrelated ones.

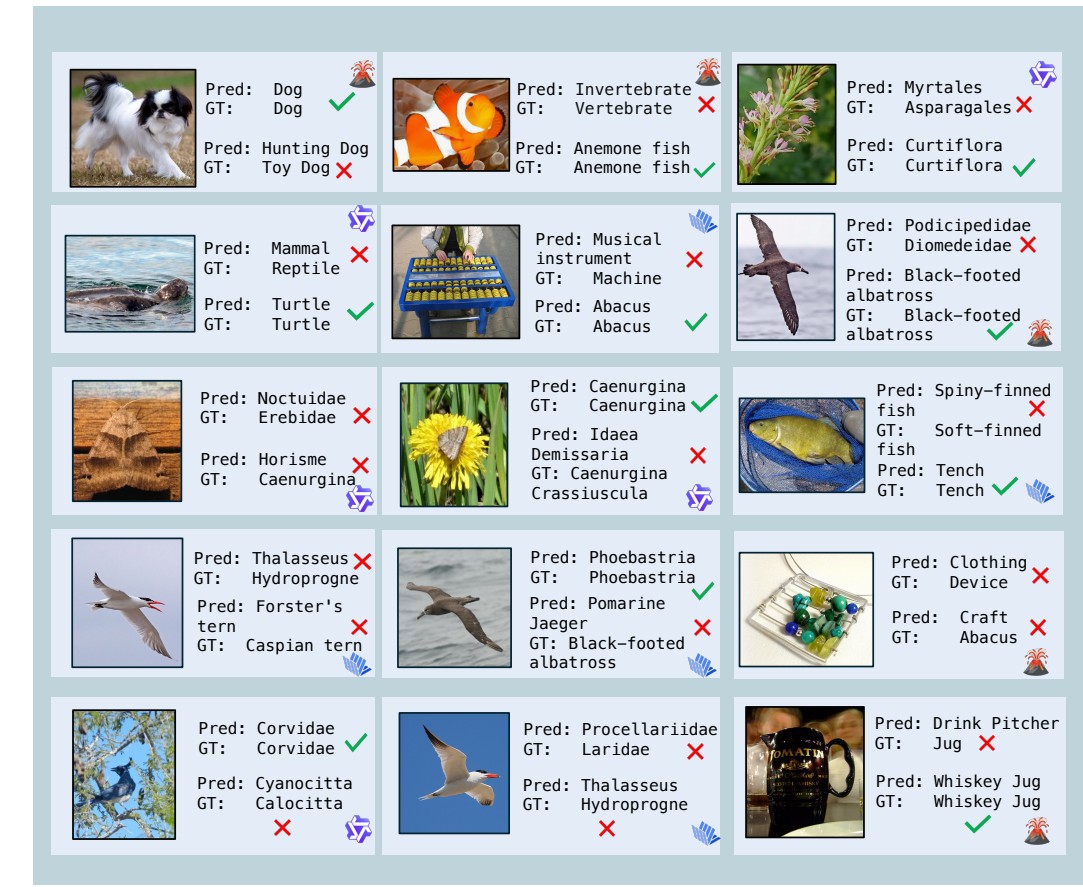

Figure 10: **Error Examples of the hierarchical predictions of VLLMs.** Examples are drawn from different VLLMs (Qwen2.5-VL-7B, InternVL2.5-8B and LLaVA-OV-7B) to reflect the diverse error modes observed across taxonomic levels.

Table 10: HCA and leaf-level accuracy $Acc_{leaf}$ of Qwen2.5-VL-7B on open-set VQA tasks across five benchmarks.

| Model | iNat21-Animal | | iNat21-Plant | | ImgNet-Artifact | | ImgNet-Animal | | CUB-200 | |
|---|---|---|---|---|---|---|---|---|---|---|
| | HCA | $Acc_{leaf}$ | HCA | $Acc_{leaf}$ | HCA | $Acc_{leaf}$ | HCA | $Acc_{leaf}$ | HCA | $Acc_{leaf}$ |
| Qwen2.5-VL-7B | 0.01 | 8.33 | 0.11 | 3.88 | N/A | 7.43 | N/A | 15.46 | 9.39 | 22.02 |

## B.6 OPEN-SET EVALUATION RESULTS

We also evaluate the open-set scenario on Qwen2.5-VL-7B (Table 10), where no answer choices are provided. In this setting, model performance drops significantly, particularly on the iNat21-Plant benchmark, where the model struggles to generate correct answers. This results in very low fine-grained accuracy and HCA.

## B.7 FOOD-101 RESULTS

A comprehensive evaluation on Food-101 with all metrics is shown in Table 11. On the Food-101 dataset, all models achieve relatively high fine-grained classification accuracy. Unlike other datasets, LLaVA-OV-7B attains the highest HCA on this benchmark among the 7B/8B open-source VLLMs, even though its leaf-level accuracy is not the highest.

Table 11: Evaluation results across all VLMs on Food-101 with all metrics reported.

| Model | $Acc_{leaf}$ | HCA | POR | S-POR | TOR |
|---|---|---|---|---|---|
| **Open-source VLLMs** | | | | | |
| LLaVA-OV-7B | 88.80 | 46.45 | 77.30 | 57.70 | 57.06 |
| InternVL2.5-8B | 84.76 | 41.18 | 72.91 | 52.77 | 51.85 |
| InternVL3-8B | 84.88 | 37.95 | 71.26 | 49.11 | 48.96 |
| Qwen2.5-VL-7B | 90.51 | 43.11 | 75.15 | 53.48 | 53.13 |
| Qwen2.5-VL-32B | 89.13 | 47.32 | 76.99 | 56.83 | 57.93 |
| Qwen2.5-VL-72B | 92.02 | 52.00 | 80.46 | 60.95 | 62.33 |
| **CLIP Models** | | | | | |
| OpenCLIP | 93.89 | 37.53 | 72.73 | 49.76 | 44.83 |
| SigLIP | 97.17 | 42.49 | 73.68 | 50.03 | 51.69 |
| BioCLIP | 27.82 | 2.48 | 26.29 | 10.81 | 7.25 |
| BioCLIP2 | 75.89 | 14.10 | 45.19 | 20.81 | 18.39 |
| **Proprietary VLLM** | | | | | |
| GPT-4o | 95.67 | 55.60 | 82.97 | 63.03 | 66.79 |

## B.8 HierarCaps Results

We further experiment with HierarCaps (Alper & Averbuch-Elor, 2024) under this work's settings, and the results are presented in Table 12. Both CLIPs and VLLMs have poor hierarchical consistency and are especially worse on abstract and shorter captions at higher levels of the hierarchy (i.e., the first two levels). This is likely due to the noisy nature of the top-level labels in the HierarCaps dataset, where each image can be associated with multiple coarse-grained captions. For example, the image with the caption "A table with three plates with food and a man and a woman both holding utensil near plates" can reasonably be classified under both "persons" and "table" categories at the first layer of the hierarchy. We also compute the HCA for the last two levels (Level 3 and Level 4), which contain more concrete and longer captions. However, there remains a noticeable gap between the HCA and the leaf node accuracy (Level 4). In many cases, the model can correctly classify at level 4 while still struggling at level 3, suggesting that the model does not fully understand the hierarchical relationship across longer textual contexts.

Table 12: Hierarchical evaluation results on HierarCaps (Level Accuracy and HCA).

| Model | Level 4 | Level 3 | Level 2 | Level 1 | HCA (All) | HCA (Last two levels) |
|---|---|---|---|---|---|---|
| OpenCLIP | 68.80 | 50.70 | 28.40 | 17.10 | 5.70 | 45.30 |
| SigLIP | 72.90 | 52.50 | 26.80 | 15.50 | 5.70 | 49.20 |
| LLaVA-OV-7B | 78.60 | 60.70 | 32.60 | 22.20 | 9.10 | 55.60 |
| Qwen2.5-VL-7B | 77.10 | 58.40 | 30.50 | 21.10 | 7.10 | 54.00 |

## B.9 Performance Analysis across Different Datasets

Among all datasets, CUB-200 exhibits the smallest gap between HCA and leaf-level accuracy, which can be attributed to its shallow hierarchy (only four levels vs. six levels in iNat21-Plant and iNat21-Animal), making the task relatively simple compared to other datasets. ImgNet-Animal and ImgNet-Artifact have the deepest hierarchies. However, their leaf nodes generally correspond to basic-level concepts, which makes the leaf-level classification easier for VLLMs, resulting in high leaf accuracy. ImgNet-Artifact is the most challenging dataset on which all models yield low hierarchical consistency scores, probably for two main reasons. (1) Unlike the well-defined biological hierarchies in iNat-21 and ImgNet-Animal, the intermediate nodes in WordNet, which were used to construct ImageNet, are relatively abstract and vague, making hierarchical discrimination difficult. (2) Unlike the animal images, images in the ImgNet-Artifact dataset often contain multiple objects of different classes. When queried about higher-level categories, the model may mistakenly associate the question with another non-central object in the scene.

### B.10 COULD THE POOR HIERARCHICAL CONSISTENCY ORIGINATE FROM THE FOUR-CHOICE VQA FORMAT?

In general, VQA benchmarks (Li et al., 2023a; Liu et al., 2024c) adopt a multiple-choice question format, with four-choice questions comprising the majority. Current open source VLLMs (Bai et al., 2025; Zhu et al., 2025; Chen et al., 2024; Li et al., 2025) have already demonstrated strong performance on these general VQA benchmarks. Therefore, the poor performance observed in our setting is unlikely to be caused by the question format or prompt design, but rather by the limitations of the VLLMs themselves. A more comprehensive analysis of the effects of prompt design and question formats on the hierarchical understanding of VLLMs is provided in Appendix C.

## C SUPPLEMENTARY MATERIALS FOR SECTION 3 IN THE MAIN PAPER

### C.1 PROMPT ENGINEERING

To comprehensively assess how prompt design affects hierarchical classification performance, we evaluate a diverse set of prompt engineering strategies.

#### C.1.1 PROMPT VARIATION

Across all benchmarks we employ five distinct prompt templates (Table 13), comprising both hierarchy-aware (Hierarchical) and general formulations (General). For CUB200, iNat21-Animal, and iNat21-Plant, we use two hierarchy-specific prompts and three general prompts. For ImgNet-Animal and ImgNet-Artifact, all five prompts are general because the corresponding taxonomy trees are highly unbalanced, making level-specific queries ill-posed. We report the results in Table 14, averaging performance separately over general (General Prompts) and hierarchy-aware prompts (Hierarchy Prompts). Overall, hierarchy-aware prompts outperform general prompts on CUB-200 and iNat21-Plant.

Table 13: Prompt templates used across datasets. Placeholders: (i) **CUB-200**: level ∈ {order, family, genus, species}; (ii) **iNat21**: object ∈ {animal, plant}, level ∈ {kingdom, phylum, class, order, family, genus, species}; (iii) **ImgNet**: class ∈ {animal, artifact}.

| Dataset | Format | Prompt Template |
|---|---|---|
| CUB-200 | Hierarchical | Based on taxonomy, what is the {level} of the bird in this image? 
 Based on the image, what is the taxonomic classification at the {level} level? |
| | General | What is the taxonomic classification of the bird in this image? 
 How can the bird in this image be categorized taxonomically? 
 What is the systematic position of the bird shown in the image? |
| iNat21 | Hierarchical | Based on taxonomy, where does the {object} in the image fall in terms of {level}? 
 Given the {object} in the image, what is its taxonomic classification at the {level} level? |
| | General | What could the {object} in the image be classified as? 
 How can the {object} in the image be taxonomically categorized? 
 What is the systematic position of the {object} in the image within the biological hierarchy? |
| ImgNet | General | What is the taxonomic category of the {class} in this image? 
 How can the {class} in this image be categorized in taxonomy? 
 Based on classification, what type of {class} is this? 
 What is the hierarchical class of the {class} shown here? 
 Where does this {class} belong in the taxonomic hierarchy? |

#### C.1.2 CHAIN OF THOUGHT (CoT)

To examine whether Chain-of-Thought reasoning improves hierarchical inference, we follow (Kojima et al., 2022; Wei et al., 2022). Concretely, we append the phrase "Let's think step by step." to the end of each question prompt followed the work (Zhang et al., 2024b). The results are presented in

Table 14: Evaluation of open-source VLLMs on hierarchical image classification benchmarks using different prompt engineering methods.

| Model | Prompt | CUB-200 | | ImgNet-Animal | | iNat21-Plant | |
|---|---|---|---|---|---|---|---|
| | | HCA | $Acc_{leaf}$ | HCA | $Acc_{leaf}$ | HCA | $Acc_{leaf}$ |
| LLaVA-OV-7B | General Prompts | 11.44 | 43.44 | 35.58 | 65.45 | 3.88 | 27.24 |
| | Hierarchy Prompts | 11.24 | 43.94 | N/A | N/A | 4.54 | 27.45 |
| | + CoT (Wei et al., 2022) | 10.99 | 42.98 | 35.72 | 65.82 | 4.46 | 27.34 |
| | + Taxonomy | 14.45 | 40.25 | N/A | N/A | N/A | N/A |
| InternVL2.5-8B | General Prompts | 19.81 | 45.58 | 37.20 | 65.12 | 4.98 | 28.11 |
| | Hierarchy Prompts | 21.25 | 45.30 | N/A | N/A | 5.61 | 28.28 |
| | + CoT (Wei et al., 2022) | 21.17 | 45.21 | 36.99 | 65.38 | 5.49 | 28.26 |
| | + Taxonomy | 16.19 | 37.88 | N/A | N/A | N/A | N/A |
| Qwen2.5-VL-7B | General Prompts | 40.78 | 65.38 | 55.33 | 79.93 | 16.15 | 40.51 |
| | Hierarchy Prompts | 43.66 | 65.54 | N/A | N/A | 17.21 | 41.49 |
| | + CoT (Wei et al., 2022) | 43.21 | 64.91 | 56.17 | 80.43 | 18.06 | 42.53 |
| | + Taxonomy | 32.52 | 52.66 | N/A | N/A | N/A | N/A |

Table 14, where no significant improvement is observed when building CoT upon the *Hierarchical Prompt*. Apart from the simple "Let's think step by step."prompt, we also evaluated a biologically grounded chain-of-thought prompting strategy on the iNat21-Plant and iNat21-Animal datasets, which feature more comprehensive and standardized taxonomies. Specifically, we incorporated the biological reasoning process directly into the system prompt on iNat21-Animal as follows:

```
You are an expert in hierarchical image classification.\n
Given an image and a multiple-choice question about a specific taxonomy level
(e.g., genus, family), first infer the most likely species from the image,
then reason step-by-step through the taxonomy hierarchy to identify the correct
label.\n
Respond with only the letter corresponding to the correct answer.\n
Example: if the image depicts Caenurgina crassiuscula, then the correct genus is
Caenurgina, family is Erebidae, order is Lepidoptera, class is Insecta, phylum
is Arthropoda, and kingdom is Animalia.\n
For instance, if the question is:\n Given the animal in the image, what is its
taxonomic classification at the phylum level?\n
A. Annelida\n
B. Arthropoda\n
C. Mollusca\n
D. Chordata\n
You should select option B, labeled with Arthropoda.
```

The hierarchical reasoning example in the system prompt for iNat21-Plant is adapted accordingly using a representative example from the iNat21-Plant taxonomy.

We report the evaluation results of Qwen2.5-VL-7B in Table 15. Notably, incorporating the biological chain-of-thought does not yield performance improvements and even underperforms compared to the simple chain-of-thought prompting strategy, as shown in Table 14.

Table 15: Biological chain-of-thought results on iNat21-Plant and iNat21-Animal using Qwen2.5-VL-7B.

| Model | iNat21-Plant | | iNat21-Animal | |
|---|---|---|---|---|
| | HCA | $Acc_{leaf}$ | HCA | $Acc_{leaf}$ |
| Qwen2.5-VL-7B | 15.42 | 40.96 | 15.39 | 40.47 |

### C.1.3 TAXONOMY AS CONTEXT

The taxonomy is encoded as a JSON dictionary that maps each leaf node to the ordered list of its ancestors up to the root. We provide this structure verbatim at the beginning of the prompt by concatenating "Here's a taxonomy: " + {Taxonomy JSON} + {original prompt}. This supplies the model with the full taxonomic context. We report results on representative open-source VLLMs using the CUB-200 dataset in Table 14. Surprisingly, explicitly providing the taxonomy as context to VLLMs does not improve performance; instead, it leads to a degradation in HCA. This may be attributed to the additional taxonomy consuming a portion of the model's attention capacity, thereby reducing the attention available for visual tokens. In addition, we include a text-only evaluation where each prompt is contextualized with the full taxonomy. The results are summarized in Table 16. Notably, even when the explicit textual taxonomy is provided, the text-only HCA reaches only 74.82%, which remains substantially below our expectations for LLMs.

Table 16: (Text) HCA of Qwen2.5-VL-7B on the CUB-200 dataset with taxonomy as context.

| LLM of | HCA | POR | S-POR | TOR |
|---|---|---|---|---|
| Qwen2.5-VL-7B | 66.26 | 89.94 | 77.08 | 77.44 |
| Qwen2.5-VL-7B w/ Taxonomy | 74.82 | 93.14 | 83.72 | 83.37 |

### C.1.4 FEW-SHOT VQA

We conducted additional experiments on the CUB-200 dataset using Qwen2.5-VL-7B with few-shot prompting ranging from 1 to 5 shots. We use level-specific QA pairs as few-shot examples to evaluate performance at each hierarchy level. An example is provided as follows:

```
Based on taxonomy, where does the <leaf label> (e.g., Black-footed Albatross)
fall in terms of <level> (e.g., order)?

A. <Ground Truth> (e.g., procellariiformes)
B. <Similar Choice> (e.g., apodiformes)
C. <Similar Choice> (e.g., podicipediformes)
D. <Similar Choice> (e.g., pelecaniformes)

Answer with the option's letter from the given choices directly.
Answer: A
```

Results are reported in Table 17. Interestingly, we observe no performance improvement across different numbers of few-shot examples.

Table 17: HCA performance across different few-shot settings.

| #Few-shot | 0 | 1 | 2 | 3 | 4 | 5 |
|---|---|---|---|---|---|---|
| HCA | 66.26 | 64.83 | 65.57 | 65.50 | 66.10 | 65.46 |

### C.1.5 QUESTIONS WITH BINARY ANSWER

We also evaluate a binary question-answering format with Yes or No responses. For each original four-choice question, we convert the four candidate answers into four separate statements. We then perform four separate forward passes on the same image to obtain the final predictions using majority voting with the results from the standard prompt. The binary-format questions are formulated as follows:

**Statement 1:** `<image>` The bird in the image belongs to the `<hierarchy>`
(e.g., Order) of `<ground truth>` (e.g., Passeriformes).
Is this statement correct? Please answer Yes or No.
**Statement 2:** `<image>` The bird in the image belongs to the `<hierarchy>`
(e.g., Order) of `<similar class>`.
Is this statement correct? Please answer Yes or No.
**Statement 3:** `<image>` The bird in the image belongs to the `<hierarchy>`
(e.g., Order) of `<similar class>`.
Is this statement correct? Please answer Yes or No.
**Statement 4:** `<image>` The bird in the image belongs to the `<hierarchy>`
(e.g., Order) of `<similar class>`.
Is this statement correct? Please answer Yes or No.

In this scenario, if none or multiple "Yes" responses appear among the four statements, we consider the model uncertain at the current hierarchy level and mark the prediction as incorrect. A prediction is counted as valid only when exactly one "Yes" answer is returned out of the four questions. Based on this criterion, we assess the answers and report the results on all metrics on CUB-200 using Qwen2.5-VL-7B in Table 18. Compared with the original four choice question answering setting, this scenario exhibits a significant performance drop, with approximately 27% degradation in HCA and 13% in leaf level accuracy. This result is expected, as the model no longer has access to contrasting choices within a single forward pass. In uncertain cases, the absence of explicit alternatives makes it more prone to errors, whereas the four choice setting can implicitly guide the model toward a correct selection by constraining the label space.

Table 18: Hierarchical evaluation results using binary QA format on CUB-200.

| Model | HCA | $\text{Acc}_{\text{leaf}}$ | POR | S-POR | TOR |
|---|---|---|---|---|---|
| Qwen2.5-VL-7B | 16.22 | 51.71 | 63.23 | 41.37 | 42.60 |

## C.2 LINEAR PROBING OF VISUAL FEATURES

For linear probing experiments on image features, we use Qwen-2.5VL-7B and retrieve image token embeddings from three checkpoints in the pipeline: (i) vision encoder output, (ii) projector output, and (iii) residual stream of the final layer of LLM. We evaluate two pooling heuristics: mean pooling across all image tokens versus selecting the final image token, and observe that mean pooling consistently outperforms the final-token alternative. Accordingly, all results in Section 3.2 are reported with mean-pooled representations, echoing the empirical findings of Zhang et al. (2024b).

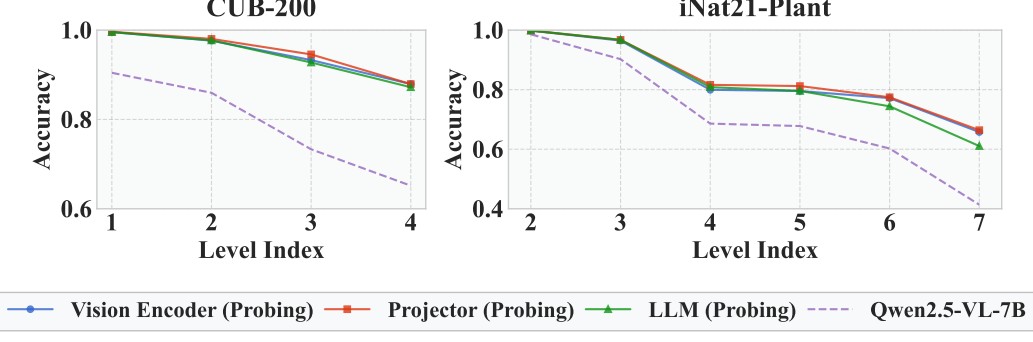

Figure 11: Level-by-level linear probing accuracy on CUB-200 (Wah et al., 2011) and iNat21-Plant (Van Horn et al., 2021) using Qwen2.5-VL-7B (Bai et al., 2025). High performance obtained from features taken at the vision encoder, vision projector, and LLM shows that discriminative visual information is preserved end-to-end throughout the VLLM.

We train a linear classifier on the training sets of CUB-200 and iNat21-Plant using a batch size of 512, a learning rate of 1e-4, and the Adam optimizer for 500 epochs. For CUB-200, we use the entire training set (5994 images), while for iNat21-Plant, we randomly sample 10 images per class to ensure a balanced subset (42710 images). For testing, we use 5794 testing images from CUB-200 and 42710 images from iNat21-Plant. Furthermore, for each level in the taxonomy, we train a separate linear classifier. After training, we report the best test performance achieved during the training process. We present the level-by-level accuracy of the probing results, as shown in Figure 11. On the iNat21-Plant dataset, we observe that the performance gap between the VLLM and the probed components increases with taxonomy depth, indicating that VLLM struggle more at finer-grained levels. In contrast, on the CUB-200 dataset, the probed components significantly outperform the VLLM across all levels. These results demonstrate that the visual embeddings are highly effective for both hierarchical consistency and fine-grained recognition. However, performance still drops when the task involves extremely fine-grained categories such as the leaf level in iNat21-Plant, which contains 4,271 distinct classes, where even the probing model achieves only 65% accuracy.

### C.3    TEXT HCA ON LARGE QWEN2.5-VLS

We also evaluated text-only hierarchical classification on the 32B and 72B variants of Qwen2.5-VL (Bai et al., 2025), with results presented in Table 19. The findings align with our observations regarding the scaling law in hierarchical classification: models with a larger number of parameters demonstrate stronger hierarchical visual understanding. However, the highest HCA across all datasets, 92.98% for Qwen2.5-VL-72B still falls short of expectations. This suggests that even with a stronger model, shallower taxonomy, and smaller dataset, the LLM's hierarchical consistency remains suboptimal. Furthermore, the consistently high Pearson correlation coefficients between text-based and visual HCAs reinforce the conclusion that the LLM component is the primary bottleneck in VLLM's hierarchical visual understanding.

Table 19: (Text) HCA of VLLMs' LLMs and its correlation $\rho$ with VLLMs' (visual) HCA on Qwen2.5-VL-32B and Qwen2.5-VL-72B.

| LLM of | iNat21-Animal | iNat21-Plant | ImgNet-Artifact | ImgNet-Animal | CUB-200 | $\rho$(text,visual) |
|---|---|---|---|---|---|---|
| Qwen2.5-VL-32B | 67.88 | 72.88 | 41.91 | 82.18 | 90.62 | 0.9517 |
| Qwen2.5-VL-72B | 83.08 | 87.76 | 41.19 | 84.51 | 92.98 | 0.9192 |

### C.4    HCA OVER DIFFERENT TAXONOMY DEPTH

To investigate which taxonomy levels contribute the most to performance degradation, we report the HCA across different taxonomy depths for both image-based and text-only hierarchical classification tasks using Qwen2.5-VL-7B, InternVL2.5-8B, and LLaVA-OV-7B on the iNat21-Plant dataset (Table 20). For VLLMs, we recompute HCA by treating upper taxonomy levels as the leaf level. For LLMs, we re-run the experiments by substituting the original leaf-node labels with higher-level labels (e.g., replacing species-level labels at level 6 with genus-level labels at level 5).

The results show that VLLMs consistently perform better as the taxonomy depth becomes shallower, which is expected since the label space decreases. However, a notable drop in performance is observed at level 5 for all models and at level 3 for Qwen2.5-VL-7B and LLaVA-OV-7B. This suggests that these specific levels of the iNat21-Plant taxonomy may represent bottlenecks for the LLMs' hierarchical reasoning capabilities.

### C.5    COMPARISON BETWEEN VISION-TUNED LLMS AND ORIGINAL LLMS

We present an extended comparison between vision-tuned LLMs and their original counterparts for all 7B/8B open-source VLLMs in Figure 12. As shown, with the exception of LLaVA-OV-7B and InternVL3-8B, all other models exhibit improved performance in their vision-tuned versions on at least 4 out of the 5 benchmarks.

Table 20: HCA of different VLLMs and their LLMs over the iNat21-Plant taxonomy of various depths.

| VLLM | Level 6 | Level 5 | Level 4 | Level 3 | Level 2 | Level 1 |
|---|---|---|---|---|---|---|
| Qwen2.5-VL-7B | 17.67 | 35.15 | 51.86 | 65.32 | 90.53 | 98.81 |
| InternVL2.5-8B | 5.66 | 14.58 | 28.35 | 43.03 | 74.82 | 90.99 |
| LLaVA-OV-7B | 4.62 | 11.83 | 25.22 | 42.30 | 78.40 | 96.12 |
| **LLM of** | Level 6 | Level 5 | Level 4 | Level 3 | Level 2 | Level 1 |
| Qwen2.5-VL-7B | 64.22 | 60.06 | 79.78 | 65.99 | 99.86 | N/A |
| InternVL2.5-8B | 41.15 | 38.38 | 65.49 | 83.76 | 99.67 | N/A |
| LLaVA-OV-7B | 28.49 | 27.95 | 55.20 | 49.46 | 99.82 | N/A |

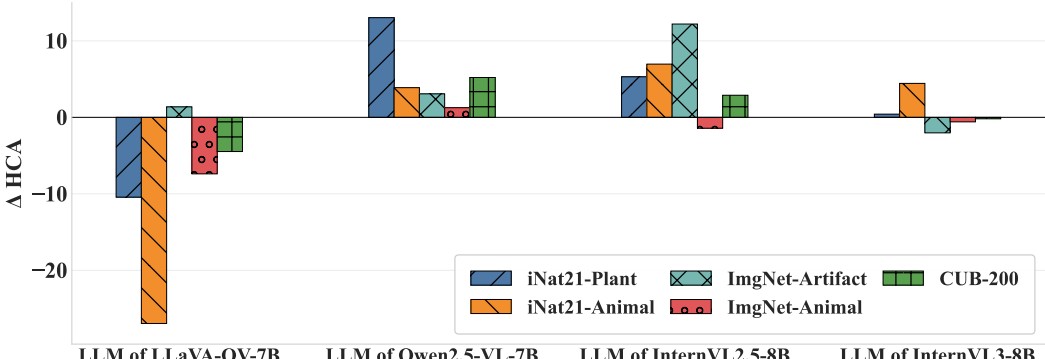

Figure 12: HCA difference between vision-tuned LLMs and their original versions across all 7B/8B open-source VLLMs. (ΔHCA = Vision-Tuned HCA - Original HCA.)

## C.6 LINEAR PROBING OF TEXT FEATURES

To quantify the extent to which hierarchical structure is preserved in the residual stream of the LLM, we perform linear probing using text token embeddings from the residual stream (across all decoder layers) of the LLM component in Qwen2.5-VL-7B, evaluated on iNat21-Plant and CUB-200. We adopt three prompt templates (listed in Table 21) that differ in semantic framing, with Prompts 1 and 2 encoding explicit hierarchical information and Prompt 3 capturing it implicitly. Following the setup in Appendix C.2, we apply mean pooling over all text token embeddings and use the same training configuration.

For text probing, we partition the taxonomy from the leaf level using an approximate 3:2 training-to-testing split ratio. This ensures that both sets share all higher-level taxonomy nodes, allowing for a unified label space across training and testing for the linear classifier. Specifically, we use 2,508 leaf nodes for training and 1,763 for testing in iNat21-Plant, and 137 leaf nodes for training and 63 for testing in CUB-200.

Table 21: Prompt templates for text probing queries. For example, {species} = Panthera leo, {hierarchy} = genus, {label} =Panthera.

| Prompt ID | Template |
|---|---|
| Prompt 1 | {species} belongs to the {hierarchy} {label}. |
| Prompt 2 | Given the {species}, what is its taxonomic classification at the {hierarchy} level? It belongs to {label}. |
| Prompt 3 | Given the {species}, what is its taxonomic classification at the {hierarchy} level? |

## C.7 Hierarchical Text Classification Results of Gemma Models

We report hierarchical text classification performance for the Gemma models (Team et al., 2024) evaluated by Park et al. (2024a) on ImgNet-Animal in Table 22, including both the 2B and 7B variants, as well as their base and instruction-tuned (IT) versions. All Gemma models perform poorly on our hierarchical benchmarks. Although the base Gemma-7B variant is the strongest among the Gemma family, it still yields the lowest text HCA compared to all other evaluated open-source VLLMs. This result suggests that even when a model exhibits perfect orthogonality in the geometric representation of hierarchical concepts, as reported in (Park et al., 2024a), it may still lack hierarchical consistency in practice.

Table 22: Hierarchical text classification performance of Gemma models on ImgNet-Animal dataset.

| Model | Gemma-2B | Gemma-2B-IT | Gemma-7B | Gemma-7B-IT |
|---|---|---|---|---|
| **HCA** | 1.11 | 16.74 | 39.57 | 29.22 |
| **POR** | 42.22 | 70.37 | 84.98 | 79.95 |

# D  Supplementary Materials for Section 4 in the Main Paper

## D.1  Training Data Construction

Following the format of hierarchical image classification benchmarks, we construct visual instruction tuning as a multi-turn question-answering task. Each question is a four-choice multiple-choice query, and each answer is a single letter denoting the correct choice, mirroring the style of the LLaVA instruction-tuning dataset (Liu et al., 2023). We adopt the **iNat21-Plant** *training* split, which contains 4,271 species (leaf nodes). Of these, we allocate 3,771 species nodes for training and hold out 500 species nodes for out-of-domain evaluation. The hierarchy distribution of the training and testing split is depicted in Figure 13. For each leaf node, we sample 10 images from the training set, yielding 37,710 training images in total. Each image is paired with a five-turn conversation that traverses the taxonomy from the class level down to the species (leaf) level. From the unused training images we construct a *validation* split by sampling 3 images per node for *all* 4,271 species, resulting in 12,813 images. This split is used for model selection and early-stopping.

## D.2  Implementation Details

During finetuning, we freeze the parameters of both the vision encoder and the vision-language projector of Qwen2.5-VL-7B, updating only the LLM component using LoRA (Hu et al., 2022) adapters. We adopt a batch size of 128 and a learning rate of $5 \times 10^{-5}$, optimized with AdamW and a warm-up ratio of 0.03. The LoRA configuration consists of a rank of 64, an $\alpha$ value of 64, and a dropout rate of 0.2. Training is performed for 1 epoch using 4 A6000 GPUs, resulting in a total of 295 steps completed within 1 hour. We report results using the model checkpoint that achieves the best performance on the validation set.

## D.3  Text-only LoRA Finetuning

Table 23: (Visual) HCA and $\text{Acc}_{\text{leaf}}$ of Qwen2.5-VL-7B before and after the (text-only) LoRA-finetuning.

| Model | iNat21-Animal | | iNat21-Plant | | ImgNet-Animal | | CUB-200 | |
|---|---|---|---|---|---|---|---|---|
| | HCA | $\text{Acc}_{\text{leaf}}$ | HCA | $\text{Acc}_{\text{leaf}}$ | HCA | $\text{Acc}_{\text{leaf}}$ | HCA | $\text{Acc}_{\text{leaf}}$ |
| Qwen2.5-VL-7B | 19.43 | 41.33 | 17.67 | 41.61 | 56.00 | 80.01 | 43.76 | 65.50 |
| Qwen2.5-VL-7B (LoRA) | 22.54 | 44.71 | 21.81 | 42.22 | 56.67 | 79.85 | 44.43 | 65.15 |
| $\Delta$ | +3.11 | +3.38 | +4.14 | +0.61 | +0.67 | -0.16 | +0.67 | -0.35 |

To investigate whether finetuning the LLM with *purely* text supervision can enrich its hierarchical representations, and thereby enhance the VLLM's hierarchical visual understanding, we create a

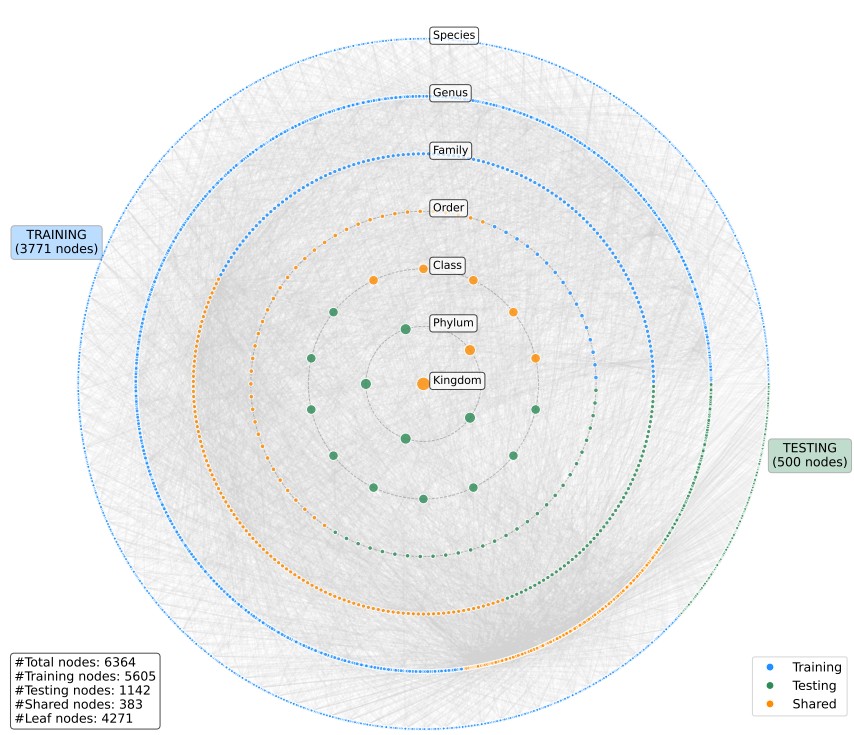

Figure 13: Hierarchy distribution of the iNat21-Plant training and testing splits.

Table 24: (Text) HCA of the LLM of Qwen2.5-VL-7B before and after the (text-only) LoRA-finetuning.

| Model | iNat21-Animal | iNat21-Plant | ImgNet-Animal | CUB-200 |
|---|---|---|---|---|
| LLM of Qwen2.5-VL-7B | 52.08 | 64.21 | 68.14 | 63.86 |
| LLM of Qwen2.5-VL-7B (LoRA) | 62.72 | 87.67 | 70.76 | 67.92 |
| Δ | +10.64 | +23.46 | +2.62 | +4.06 |

*text-only* instruction-tuning corpus. Similar to what we did in text-only hierachical benchmark curation, this dataset is obtained by replacing the image tokens from our visual instruction-tuning corpus by the leaf node label while preserving the multi-turn prompts and their ground-truth answers. For the text-only finetuning, we adopt the same training setup as described in Appendix D.2.

The evaluation results on hierarchical VQA benchmarks are shown in Table 23, and the corresponding results on hierarchical text-only QA benchmarks are presented in Table 24. As seen in Table 23, although the improvements are modest, the model shows consistent gains in the four evaluated benchmarks, with an increase of 4.14 in HCA on iNat21-Plant and 3.11 on iNat21-Animal. This suggests that enhancing the hierarchical understanding of LLM in the language space can also benefit the hierarchical visual reasoning of VLLM, reinforcing our earlier finding that the LLM component is a key bottleneck.

For the text-only results in Table 24, the model achieves performance that is, on average, comparable to the vision instruction-tuned model. Notably, the performance gains on iNat21-Plant and CUB-200 exceed those of the vision-tuned model (Table 4), whereas the improvements are smaller on iNat21-Animal and ImgNet-Animal.

### D.4 EVALUATION ON GENERAL VQA BENCHMARKS

We report the evaluation results of our vision instruction-tuned model on three general VQA benchmarks: MME (Fu et al., 2023), MMBench (Liu et al., 2024c), and SEED-Bench (Li et al., 2023a), as shown in Table 25. Notably, our hierarchically enhanced VLLM demonstrates no degradation in general-purpose performance and even achieves improvements on MME and MMBench. These results suggest that our finetuned model can serve both as a specialized assistant for users interested in taxonomy and as a general-purpose VLLM for broader applications.

Table 25: Performance comparison between the original Qwen2.5-VL-7B (OG) and our (vision) LoRA-tuned variant (LoRA) on three general VLLM benchmarks.

| Model | MME | MMBench | SEED-Bench |
|-------|-----|---------|------------|
| OG | 2306 | 82.04 | **75.95** |
| LoRA | **2345** | **83.25** | 75.93 |

We also report results for the text instruction-tuned model in Table 26. Consistent with the vision instruction-tuned performance, we observe no loss of generalization ability. This further confirms that our hierarchical fine-tuning datasets are helpful and can be seamlessly integrated into both VLLM and LLM instruction-tuning pipelines.

Table 26: Performance comparison between the original Qwen2.5-VL-7B (OG) and our (text) LoRA-tuned variant (LoRA) on three general VLLM benchmarks.

| Model | MME | MMBench | SEED-Bench |
|-------|-----|---------|------------|
| OG | 2306 | 82.04 | **75.95** |
| LoRA | **2357** | **82.39** | 75.84 |

## E  DETAILED DISCUSSION OF RELATED WORKS

**Hierarchical classification.** Hierarchical classification (Silla & Freitas, 2011; Kosmopoulos et al., 2015) involves assigning labels from a structured semantic hierarchy rather than from a flat label space lacking relational structure. In the vision domain, hierarchical image classification aims to improve visual consistency across coarse-to-fine categories, thereby enhancing overall classification performance. Recent work has introduced structural priors into visual models through hierarchical loss functions, multi-level supervision, and taxonomy-aligned embeddings (Yi et al., 2022; Park et al., 2024b; Zeng et al., 2024; Sinha et al., 2024; Chen et al., 2022). Beyond the visual domain, hierarchical classification has also been extensively explored in the language domain (Zhou et al., 2020; Wang et al., 2022; Zhou et al., 2025). Similar to approaches developed for enhancing hierarchical consistency in vision models, prior work has focused on injecting hierarchical information into language encoders to improve the structure-awareness of text embeddings. He et al. (2024) retrained transformer-based language models in hyperbolic space, resulting in improved modeling of hierarchical knowledge. Another line of research aims to understand how hierarchical structures are inherently encoded within pre-trained language models. Nikishina et al. (2023) provide a comprehensive analysis of transformer-based models for the task of hypernymy prediction, evaluating their ability to infer IS-A relations. Lin & Ng (2022) study whether pre-trained BERT models capture the transitivity of IS-A relations in WordNet. To more rigorously assess such capabilities, He et al. (2023) propose ONTOLAMA, a benchmark and evaluation framework targeting subsumption inference within ontologies. Similarly, Moskvoretskii et al. (2024) evaluate the WordNet-based lexical-semantic reasoning ability of the LLaMA-2-7B model through the TaxoLLaMA framework.

**Hierarchical classification with VLMs.** Existing studies have shown that CLIP models (Radford et al., 2021) struggle to maintain semantic consistency across taxonomic levels (Wu et al., 2024; Xia et al., 2023; Pal et al., 2024; Geng et al., 2023). ProTect (Wu et al., 2024) evaluated the CLIP model across different levels of semantic granularity and proposed a hierarchy-consistent prompt tuning method. HyCoCLIP (Pal et al., 2024) leveraged the inherent hierarchical nature of hyperbolic

embeddings to improve the hierarchical structuring of CLIP representations. HGCLIP (Xia et al., 2023) further advanced this direction by combining CLIP with graph-based representation learning to better exploit the hierarchical class structure. By leveraging the hierarchy information, CHiLS (Novack et al., 2023) improves the zero-shot classification accuracy of the CLIP model.

**Classification with VLMs.** While VLMs (Bai et al., 2025; Zhu et al., 2025; Li et al., 2025; Chen et al., 2024) have demonstrated strong performance across a wide range of tasks, their effectiveness in visual classification, particularly for fine-grained and subordinate-level recognition—remains suboptimal (Zhang et al., 2024b; Liu et al., 2024b; He et al., 2025; Yu et al., 2025; Conti et al., 2025; Yu et al., 2025). Zhang et al. (2024b) identified the limitations of current VLLMs in classification tasks and introduced ImageWikiQA, a new benchmark focused on object recognition. Building on this, Liu et al. (2024b) evaluated a broader range of recent VLLMs, highlighting that models such as Qwen2-VL have achieved notable improvements in classification accuracy, largely due to language model advances and the use of more diverse training data. He et al. (2025) further investigated the causes of poor fine-grained classification performance, attributing it primarily to the absence of sufficient category names during training. To better assess the classification capabilities of vision-language models, Geigle et al. (2024) proposed FOCI, a benchmark derived from five popular classification datasets. Yu et al. (2025) introduced a comprehensive fine-grained classification benchmark and demonstrated that the performance of VLLMs steadily declines as category granularity becomes finer. Beyond closed-set evaluation, Conti et al. (2025) explored the open-world classification abilities of VLLMs from a broader perspective. To better evaluate the VLLM in an open-ended format, Snæbjarnarson et al. (2025) proposed to evaluate the unconstrained text predictions in a taxonomy manner instead of the exact string matching. In contrast to previous work, we provide a more comprehensive evaluation of classification ability across different levels of semantic abstraction, enabling a finer analysis of hierarchical consistency in VLLMs.

# F  LIMITATIONS

While we have identified that the bottleneck in VLLM's hierarchical visual understanding lies in the LLM component, the underlying cause of LLMs' lack of hierarchical consistency in the language space remains an open question. Given the vast, highly structured corpora used during pre-training, one might expect stronger hierarchical representations to emerge from LLMs naturally. Unfortunately, our computational budget precludes training an LLM from scratch to verify this hypothesis. We, therefore, leave to future work the investigation of pre-training strategies that inject *explicit* hierarchical knowledge, an avenue that could clarify the underlying cause and potentially close the remaining performance gap.

Moreover, our study focused on hierarchical image classification due to limited resources. However, hierarchical visual understanding is broader, including video, 3D, and other visual modalities and more diverse taxonomies. We conjecture that state-of-the-art VLLMs would still perform poorly in those scenarios, but the causes could be different from our findings. LLMs probably would remain the weak point in those scenarios, and it is possible that the visual encoder or projector would be equally responsible.

Finally, we made a bold hypothesis that one cannot make VLLMs understand visual concepts fully hierarchical until LLMs possess corresponding taxonomy knowledge. It could overly blame LLMs, although we have supported this hypothesis with a systematic empirical investigation and the strong correlations between LLMs' taxonomy knowledge and the corresponding VLLMs' hierarchical visual understanding performance. Some post-training and test-time computation methods could work well without explicitly improving LLMs' taxonomy knowledge.

# G  BROADER IMPACTS

Accurate hierarchical visual reasoning is critical in applications where coarse- and fine-grained decisions coexist-e.g., biodiversity monitoring, medical diagnostics, autonomous driving, and content moderation. Our study uncovers a systematic weakness in current VLLMs: they often predict plausible fine-grained labels while violating higher-level taxonomic structure. Deploying such models without qualification could, for example, mislead ecological surveys, propagate medical mis-triaging, or bias downstream decision-making pipelines that rely on hierarchical consistency for error checking.

By pinpointing the LLM component as the bottleneck in VLLM's hierarchical visual understanding and demonstrating that modest multimodal finetuning already improves textual taxonomy knowledge, our findings encourage the community to (i) incorporate explicit hierarchical objectives during LLM pre-training, (ii) curate multimodal corpora with reliable taxonomic annotations, and (iii) develop evaluation metrics that penalize hierarchical inconsistency. These steps could yield models that are safer and more trustworthy in real-world, hierarchy-rich settings.

Potential downsides include the amplification of existing taxonomic biases or misclassifications if the training data encodes culturally or scientifically outdated hierarchies. Researchers should therefore audit hierarchies for regional or disciplinary bias, publish data-curation protocols, and, where feasible, provide mechanisms for community feedback and correction.

Overall, we believe that exposing and remedying hierarchical blind spots in VLLMs will enable more reliable AI systems and support scientific, environmental, and industrial domains that depend on structured semantic understanding.

## H  LLM USAGE STATEMENT

We utilize LLMs to refine the taxonomies (Appendix A.2) and convert structured classification tasks into free-form VQAs (Section 3.2). Appendix A.1 details how we use LLMs in the data curation process. Besides, we evaluate some open-source LLMs in Section 3.3 along with the corresponding VLLMs. Finally, we use proprietary LLMs to polish our writing.

