# OpenReview forum: "The LLM Bottleneck: Why Open-Source Vision LLMs Struggle with Hierarchical Visual Understanding"
_ICLR.cc/2026/Conference — ICLR 2026 Conference Withdrawn Submission_

### Official Review · Reviewer_gYiW · 2025-10-27

**Soundness:** 3
**Presentation:** 3
**Contribution:** 2
**Rating:** 4
**Confidence:** 4

**Summary:**

This paper focuses on the hierarchical visual understanding of VLLMs. By transforming existing datasets into hierarchical VQA datasets, the authors have identified that VLLMs perform badly when they need to classify a certain object at all different levels. The authors further explore the possible reasons for the bad performance by examining different components of VLLMs. The results show VLLMs’ LLMs may be the bottleneck in the open-sourced VLLMs' hierarchical visual understanding.

**Strengths:**

- The paper is well-structured and easy to understand.
- The processed hierarchical VQA benchmark and the corresponding metrics are useful to evaluate VLLMs.
- Broad tests of multiple types of models, including open-source VLLMs, CLIP models and proprietary VLLM show that hierarchical visual understanding and fine-grained visual recognition are generally challenging for those models.

**Weaknesses:**

- The related work only includes relevant visual models and has a limited introduction about other domains. Specifically, one of the key findings in this paper is that the LLM backbone is bad at hierarchical understanding. However, the related literature is not introduced. The reviewer wonders if that has been shown already by the LLM literature or scientific LLMs, which might weaken the contribution of this paper.
- On the CUB-200 subset, GPT-4o is used both as an annotator and predictor, making the results less convincing. Meanwhile, if those labels are generated by GPT-4o, why is GPT-4o's performance 98.81 (Table 2) instead of 100?
- The authors claimed that the LLM part is the bottleneck in hierarchical visual understanding for open-source VLLMs, which may not be general enough. For example, in Table 19, Qwen2.5-VL-72B also has very high HCA and starts to be closer to GPT's performance. And the recently proposed GPT-oss models may have even closer performance. The reviewer hence feels that the LLM part's bottleneck may only exist in small models, where the pretraining stage mainly focuses on common words and world knowledge, and has less capacity in memorizing scientific words/knowledge.
- The finetuned results show that LLM's text HCA (Table 4) improves more compared to Visual HCA (Table 3), which also shows that the LLM part is not the only reason causing the low performance in hierarchical understanding. However, the vision part's effect is not further explored by the authors.
Overall, the reviewer feels this paper mainly blames LLM's limitation in hierarchical understanding (which may not be fully correct given bigger LLMs' good performance and may have been studied already by LLM literature), and has not fully explored the limitations from VLLM itself.

**Questions:**

- The proposed Pearson’s correlation coefficient only shows the correlation at the overall statistical level and hard to evaluate if the bad performance only comes from LLM. One thing that could be tested is to evaluate the accuracy of visual HCA for the samples that the VLLMs answer correctly in the pure text case.

---

### Official Review · Reviewer_NFwf · 2025-11-01

**Soundness:** 3
**Presentation:** 4
**Contribution:** 3
**Rating:** 4
**Confidence:** 3

**Summary:**

This paper examines hierarchical visual understanding in Vision Large Language Models (VLLMs) and argues that the LLM is the primary bottleneck, rather than the vision encoder. The authors introduce a new large-scale benchmark of approximately one million four-choice Visual Question Answering (VQA) tasks, derived from 6 taxonomies (CUB-200, iNat-Plant/Animal, ImageNet-Animal/Artifact, Food-101) using 4 datasets.  Their key finding is that VLLMs exhibit low "Hierarchical Consistent Accuracy (HCA)," meaning they often fail to make predictions that are consistent across different levels of a taxonomy. Through a series of experiments, the authors isolate different components of the VLLM architecture (vision or text) over different layers and argue that the visual encoders and projectors are not the problem. Instead, they identify the LLM as the primary bottleneck, showcasing that they lack the necessary taxonomic knowledge to reason hierarchically.

**Strengths:**

- While prior work has noted VLLMs' struggles with fine-grained classification, the paper presents interesting insights and experiments to systematically diagnose the source of the failure in the context of hierarchical classification.

- The paper's main contribution lies in its rigorous diagnostic framework. The authors evaluate both the visual and textual components of the VLLMs independently across different layers and projection heads to give a better understanding of how each layer affects.

- Linear probing of frozen visual features achieves strong leaf accuracy and HCA for some cases, highlighting that they contain some hierarchical information

- The findings of visual encoders capturing the hierarchical information and LLM effectively encoding the hierarchical information are interesting, as visual backbones are often thought of as the weaker modality

**Weaknesses:**

- Insufficient details on the benchmark curation: The manuscript and abstract have highlighted the creation of 1 million VQA pairs; however, the details provided on how they got to the 1 million number are sparse in the main paper and only have minimal descriptions of the dataset creation in the appendix.

- Ambiguity in the "Bottleneck" Conclusion: Section 3.3.1 talks about LLMs being the bottleneck, which is the main premise of the paper. Comparing the results of the text (table 2) vs visual probing (section 3.2, figure 3), for the iNat21-Plant dataset, visual probing has the same/similar results, having an HCA of ~62% (from figure 3) as the (Text) HCA of 64.21% (table 2) for the Qwen2.5-VL-7B model. How can you conclude that the visual encoder is not the bottleneck and the LLM is if the two perform equally well? The results on CUB are better, but don’t carry over to the iNat21-Plant dataset. A similar phenomenon is seen for GPT-4o, where the text HCA is very high.

- Similarly, in Section 3.3.2, Line 377 mentions “original LLMs are on par with or even worse than their vision-tuned counterparts,” but Figure 5 shows it is better for one model and worse for the other. So how can this conclusion be drawn that they are “at par or worse”?

- Section 4, lines 422-423, emphasizes that the shortcomings of hierarchical classification can only be fixed in the pretraining stage. Whereas, in the paper, the authors show that through LoRA -finetuning (Tables 3&4) can lead to better HCA and still not have catastrophic forgetting over the other benchmarks (Tables 25 & 26).

- The discussion of related work is largely in the appendix. Integrating a more comprehensive literature review into the main paper would better contextualize the paper's contributions with respect to recent advances in hierarchical classification using VLLMs.

**Questions:**

- Choice of “confusing labels”. Line 156 mentions the use of SigLIP to compute the cosine similarity to get the closest top three most similar labels. What was the idea behind this design choice? One could have leveraged the inherent taxonomy available and gotten the top 3 closest siblings to the current class being evaluated. This would have created a more accurate “closest or confusing labels” as one could have selected a different set of species belonging to the same “Genus” level for maximum confusion.

- Highlighted in Section 2.2 (Line 161) and Section A.2, the authors use GPT-4o to create and refine the taxonomy for curating the dataset used in the paper. The paper’s goal is to evaluate the hierarchical classification accuracy, and Table 1 shows that GPT-4o is not perfect. It is contradictory that GPT-4o is used to map each class into 4 taxonomic levels (section 2.2) and use it to find inconsistencies in the GT taxonomy (Section A.2)

- Section 2.4 notes that the scaling laws of VLMs have better HCA accuracy with increasing the model size of Qwen2.5-VL. Does this hold true for other models like InternVL3 as well?

- Section C.3 line 1370 discusses that the highest Text HCA being 92.98% for Qwen2.5-VL-72B "still falls short of expectations." This is a strong statement. Could the authors please clarify what the expectations are for this metric?  Linear probing of visual features also led to ~80-85% HCA on the CUB dataset.

- Section C.5 compares between Vision-tuned LLMs and original LLMs, and Figure 14 shows that only 50% of the LLMs get better with vision-tuning.

- Clarity in Section 3.1 and Figure 2, which highlights the effect of prompting the VLLMs. Looking at the figure, it seems like the numbers are exactly the same. While the figure looks nice, it is hard to identify if there is any difference in the results of prompting, and Table 14 provides a clearer result where Hierarchical prompts have slightly better performance than general prompts

- Section C.6 discusses linear probing for Text, but does not have any results

- Lines 334 & 335 say “correspondingly, the leaf nodes are less fine-grained”. Are we talking about parent nodes being less fine-grained or leaf nodes being “more” fine-grained?

**Details Of Ethics Concerns:**

No ethical concerns.

---

### Official Review · Reviewer_T1eK · 2025-11-01

**Soundness:** 3
**Presentation:** 3
**Contribution:** 2
**Rating:** 6
**Confidence:** 3

**Summary:**

This paper investigates why open-source vision-language models (VLLMs) fail to reason consistently across hierarchical visual categories. It introduces large-scale hierarchical visual question-answering benchmarks derived from taxonomic datasets to evaluate both fine-grained and ancestor-level predictions. Through systematic experiments, the authors show that while visual encoders preserve rich hierarchical features, the underlying language models lack taxonomy knowledge and is probably the main bottleneck. They analyze this through text-only prompts, concluding that improving hierarchical visual understanding will require LLMs with explicit, structured semantic knowledge.

**Strengths:**

1.	Extensive analysis on various components of VLMs to get closer to identifying the root cause of the lack of knowledge of hierarchical relationship between objects
2.	Pinpoints the likely cause to be the inherent lack of knowledge in the LLMs by eliminating other possible reasons through various experiments. (Although this may require future work to confirm for sure.)

**Weaknesses:**

1.	More than half of the datasets used are in the biological domain. Wish there were more datasets from other domains. Especially day to day objects, which were more likely to be part of the dataset used for VLM training.
2.	The ImgNet-Artifact dataset could have been explored more. Day to day objects can belong to multiple super categories at the same time. Considering that  ImgNet-Artifact is the only dataset of such objects, pictures that show the taxonomy (a part of it since the tree is large), would have been helpful to get a feel of the dataset.
3.	Lack of prompt tuning on larger models. As discussed in [A], prompting related improvements generally show on as the model size scales. Prompting the larger models would have helped to understand if the decoding ability of larger LLMs can be improved with prompting.
4.	Handling cases where there is ambiguity in super class. The introduced metrics do not seem to be capable of handling cases where multiple super class can be suitable to particular sub-class. This is generally noticeable in day-to-day objects. For example, a drone is both a robot and an aircraft. The authors could have discussed approaches for how to handle this, and suitable metrics for this.

[A] Wang, X. and Zhou, D., 2024. Chain-of-thought reasoning without prompting. Advances in Neural Information Processing Systems, 37, pp.66383-66409.

**Questions:**

1.	There is mention of prompting GPT-4o to get the taxonomic labels  of CUB-200 and also for cleaning the hierarchy of ImgNet-Artifact. If so, then how was GPT-4o evaluated on it if ground truth is provided by the model itself?

---

### Official Review · Reviewer_ykC4 · 2025-11-01

**Soundness:** 2
**Presentation:** 3
**Contribution:** 2
**Rating:** 2
**Confidence:** 4

**Summary:**

The paper evaluates a range of vision large language models (VLLMs) on hierarchical classification tasks. Specifically, the authors introduce four-choice VQA benchmarks derived from several hierarchical classification datasets. The results indicate that current VLLMs produce poor consistency across taxonomic levels. The authors further probe visual and textual representations and conclude that the core limitation lies in the LLM component. Through fine-tuning with hierarchical information, the model (especially the LLM component) receives certain amounts of performance improvement.

**Strengths:**

•	The paper presents comprehensive empirical evaluations across multiple datasets, domains, and model families. The empirical discovery is valuable.
•	The proving analysis is well-motivated and carefully designed, and the observation that visual features retain discriminative structure across layers is convincing.
•	The insight that fine-tuning improves the LLM’s hierarchical consistency more than the VLLM’s performance is interesting and will be useful for the community.

**Weaknesses:**

•	The paper claims to study hierarchical visual understanding of VLLMs. The initial claim of understanding WHY open source LLMs perform poorly is not supported essentially by the rest of the paper and the paper plays fast and loose with the term “understanding”.  Hierarchical classification does not equal hierarchical visual understanding. Hierarchical visual understanding refers to understanding the object and its components, the ability to reason about and attribute its various properties to the different layers of the hierarchy. In the context of biology, it can be the classification of an object based on the understanding of its traits. In contrast, this paper evaluates the hierarchical visual understanding through prediction correctness in a taxonomy graph. This task generally does not involve “understanding,” as there is no reasoning or causality involved in the task design. The multi-choice classification is more about whether the model remembers this knowledge or not. A biology, or a domain-based, evaluation task might be better suited for “understanding” claims. For example, VLM4Bio: A Benchmark Dataset to Evaluate Pretrained Vision-Language Models for Trait Discovery from Biological Images
https://neurips.cc/virtual/2024/poster/97668, https://arxiv.org/abs/2408.16176

•	On the other hand, the taxonomic consistency might not be well represented in the pre-training data of LLMs, especially those related to biological taxonomy. It is unsurprising that LLMs become the limiting factor when the corresponding knowledge is limited during their pre-training. Although the evaluation is comprehensive to prove this point, it provides limited insights to readers. The lack of consistency of vanilla models has been repeatedly shown in the past, including in the context of biological taxonomies
•	Based on the above two points and also acknowledged in the paper, the designed tasks can potentially be solved by pre-training LLMs with corresponding taxonomic knowledge. This further contradicts the topic of “hierarchical visual understanding,” where vision does not influence the results. The paper fails to demonstrate whether the model distinguishes different taxonomic levels through visual cues or purely remembers label relationships. Also, improving LLM taxonomy knowledge does not necessarily lead to improved hierarchical visual understanding.

Minor:
•	From the very first paragraph, there is lack of precision and confusion in stating the biological taxonomy terms. In the context of biological taxonomy, “Boston Terrier” and “terrier” are breeds, not a class (the class is Mammalia, or mammal, with the dog, or Canis familiaris being the species). (also, typo: P.4, l.183: “Specie” should be “Species”). The full biological taxonomy of a domestic dog is below.

•	Kingdom:	•	Animalia

•	Phylum:	•	Chordata

•	Class:	•	Mammalia

•	Order:	•	Carnivora

•	Suborder:	•	Caniformia

•	Family:	•	Canidae

•	Subfamily:	•	Caninae

•	Genus:	•	Canis

•	Species:	•	C. familiaris

•	The use of “LLM” in Figure 3 and Section 3.2 is misleading. By only looking at the figure, readers might think that LLM refers to textual embeddings.

**Questions:**

Given that the models generally have higher leaf-level accuracy, could a model achieve high consistency scores simply by retrieving taxonomic levels without looking at the image? If so, does the proposed benchmark still meaningfully evaluate visual understanding?

---

### Note · Authors · 2025-11-12

I have read and agree with the venue's withdrawal policy on behalf of myself and my co-authors.